# TinyV: Reducing False Negatives in Verification Improves RL for LLM Reasoning

**Zhangchen Xu**[*]                                    *zxu9@uw.edu*
*University of Washington*

**Yuetai Li**[*]                                         *yuetaili@uw.edu*
*University of Washington*

**Fengqing Jiang**                                      *fengqing@uw.edu*
*University of Washington*

**Bhaskar Ramasubramanian**                             *bhaskar@wwu.edu*
*Western Washington University*

**Luyao Niu**                                           *luyaoniu@uw.edu*
*University of Washington*

**Bill Yuchen Lin**[‡]                                  *byuchen@uw.edu*
*University of Washington*

**Radha Poovendran**[‡]                                 *rp3@uw.edu*
*University of Washington*

**Reviewed on OpenReview:** *https://openreview.net/forum?id=HMGsqApBM3*

## Abstract

Reinforcement Learning (RL) has become a powerful tool for enhancing the reasoning abilities of large language models (LLMs) by optimizing their policies with reward signals. Yet, RL's success relies on the reliability of rewards, which are provided by verifiers. In this paper, we expose and analyze a widespread problem—false negatives—where verifiers incorrectly reject correct model outputs. Our in-depth study of the Big-Math-RL-Verified dataset reveals that over 38% of model-generated responses suffer from false negatives in math reasoning tasks, where the verifier fails to recognize correct answers. We show, both empirically and theoretically, that these false negatives severely impair RL training by depriving the model of informative gradient signals and slowing convergence.

To mitigate this, we propose TɪɴʏV, a lightweight LLM-based verifier that augments rule-based methods with a compact LLM to dynamically detect false negatives and recover valid trajectories in rule-based final-answer verification settings. Across multiple math-reasoning benchmarks, integrating TɪɴʏV improves final model performance by up to 10%, and reaches the peak performance of the rule-based verifier using fewer than 50% of the training steps. Our findings highlight the critical importance of addressing verifier false negatives in RL for verifiable reasoning tasks, particularly in math domains. Our code is available at: https://github.com/uw-nsl/TinyV.

## 1 Introduction

---

[*]These authors contributed equally to this work. [‡]Equal advising.

Reinforcement Learning (RL) has become a cornerstone for advancing the reasoning capabilities of large language models (LLMs) (Chen et al., 2025b), as evidenced by state-of-the-art models like OpenAI o1 (Jaech et al., 2024) and DeepSeek-R1 (DeepSeek-AI et al., 2025). The effectiveness of RL depends on verifiable rewards, which provide essential supervision signals for policy optimization (Lambert et al., 2024). In mathematical reasoning tasks, prior work has predominantly relied on **rule-based** verifiers (Zeng et al., 2025; Luo et al., 2025; Yang et al., 2024), which assign a binary reward by comparing the model's generated answer with the ground truth, yielding a reward of 1 if they are equivalent and 0 otherwise.

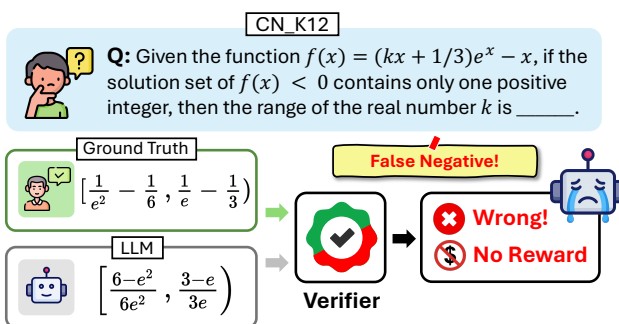

Figure 1: This figure illustrates a false negative case in the CN_K12 dataset, where the ground truth and the response generated by LLM (DeepSeek-R1-Distill-Qwen-7B) are mathematically equivalent, yet *Prime Verifier* and *Math Verify* incorrectly mark the response as wrong.

Despite the widespread use of verifiers to assess model outputs (Chen et al., 2025a; Contributors, 2023; Gao et al., 2024), their reliability in the context of RL training and its impact on performance remain underexplored. In this paper, we investigate the prevalence of **false negatives (FNs)** in answer verification, where conventional approaches (e.g., rule-based verifiers relying on string matching (Yu et al., 2025) or advanced parsing (Cui et al., 2025; Hugging Face, 2025)) fail to recognize correct answers, leading to incorrect reward assignment. Figure 1 illustrates a case where the rule-based verifiers *Prime Verifier* (Cui et al., 2025) and *Math Verify* (Hugging Face, 2025) fails to verify an equivalent answer due to their rule-based matching criteria. To quantify these issues, our analysis of the *Big-Math-RL-Verified* dataset (Albalak et al., 2025) revealed that among responses marked as incorrect by *Prime Verifier*, **38.5%** were actually correct, indicating a high prevalence of FNs. Our further analysis identified **natural language** elements in either the model's response or the ground truth answer as the primary cause of these false negatives, underscoring a critical limitation of rule-based verifiers.

The reliance on rule-based verifiers with high FNs in RL for reasoning tasks poses significant challenges for advancing research and model development. First, problems that are harder to verify using rule-based approaches, such as those involving natural language elements or complex latex expressions, are often excluded from training and evaluation, thereby limiting the model's reasoning capabilities and hindering understanding of such challenging reasoning problems. Second, the high prevalence of FNs caused by rule-based verifiers reduces training efficiency by introducing incorrect reward signals, which can mislead policy optimization and slow convergence, ultimately impeding progress in developing more robust and generalizable reasoning models.

In this paper, we investigate the impact of false negatives on RL training for mathematical reasoning through both empirical and theoretical analyses. Empirically, we find that FNs, arising from incorrect reward signals, significantly impair training efficiency by reducing the availability of informative gradient signals, particularly during early training stages. Furthermore, our theoretical analysis demonstrates that FNs hinder learnability, as measured by the reverse Kullback-Leibler (KL) divergence between policies at consecutive optimization steps, thereby slowing convergence across math reasoning benchmarks.

To address the issue of FNs in RL, we propose TinyV, a lightweight LLM-based verifier designed to enhance reward accuracy while maintaining computational efficiency. By augmenting rule-based verifiers like *Prime Verifier*, TinyV corrects FNs, enabling more effective RL training for mathematical reasoning tasks. To evaluate its performance and bridge the gap in existing benchmarks, we develop the *HardVerify-Math Bench*, which focuses on challenging verification scenarios. Our experimental results demonstrate that TinyV achieves up to a 10% improvement in pass rates across *HardVerify-Math*, with notable increases in performance on other benchmarks such as MATH and Olympiad Bench, and accelerates convergence compared to baseline verifiers. Interestingly, we found that training on questions with easily verifiable answers leads to poor performance on hard-to-verify questions, opening future research on developing more accurate reward assignment and diverse training datasets to address these challenges.

The paper is organized as follows. Section 3 introduces the preliminaries for our study. Section 4 investigates the **prevalence** and underlying causes of false negatives (FNs) in real-world RL training data. Section 5 analyzes the **impact** of FNs on reinforcement learning, from both empirical and theoretical perspectives. Section 6 presents TinyV, detailing its data curation, model training, and demonstrating how it **mitigates** false negatives to improve RL training. Limitations and ethical considerations are provided in Appendix F.

## 2    Related Work

**Rule-based Answer Verification in LLMs.** Rule-based answer verification is widely used in LLM data pre-processing (Xiong et al., 2025), model training (Yu et al., 2025; DeepSeek-AI et al., 2025; Shao et al., 2024), and evaluation frameworks such as LM Eval Harness (Gao et al., 2024), OpenCompass (Contributors, 2023), openai-evals (OpenAI, 2025), and UltraEval (He et al., 2024b). This approach assesses the correctness of LLM outputs by comparing them against ground-truth answers associated with specific datasets. However, rule-based verification may struggle to evaluate semantically equivalent but textually distinct responses, potentially resulting in false negatives (Chen et al., 2025a).

**LLM as a Judge**. The increasing capabilities of LLMs have spurred interest in using them as judges to evaluate other models, often referred to as "LLM as a judge" (Gu et al., 2025). This approach leverages LLMs' understanding to assess output quality, particularly for subjective or complex tasks where traditional metrics may fall short. LLM-as-a-judge methods are widely employed in alignment tasks (Lin et al., 2024; Li et al., 2024; Dubois et al., 2024; Li et al., 2023; Gu et al., 2025; Li et al., 2025). Recently, xVerify introduced a compact LLM as an efficient answer verifier for reasoning model evaluations, surpassing GPT-4o in overall performance (Chen et al., 2025a). Additionally, LLM-as-a-judge techniques are increasingly integrated into training processes. For instance, SEED-THINKING-V1.5 employs a reasoning model to evaluate a diverse set of verifiable questions across varied scenarios (Shao et al., 2024). Recently, Ma et al. (2025) utilizes a model-based verifier to deliver robust and accurate cross-domain rewards for RL training, and Huang et al. (2025) investigates reliable designs of verifiers in mathematical reasoning.

**Increasing Efficiency in RL for LLMs.** Recent efforts have focused on improving the efficiency of RL training for LLMs, particularly with GRPO (Shao et al., 2024). DAPO (Yu et al., 2025) enhances GRPO's efficiency by introducing dynamic sampling, which filters out prompts with accuracy values of 0 or 1, retaining only those with effective gradients while maintaining a consistent batch size. VAPO (Yuan et al., 2025b) improves the utilization efficiency of positive samples during RL training through the Positive Example LM Loss. Additionally, PODS (Xu et al., 2025) proposes max-variance down-sampling to select rollouts with maximally diverse reward signals, achieving greater efficiency compared to the GRPO baseline.

## 3    Preliminaries

**Reinforcement Learning in Language Models.** RL in the context of language models involves optimizing a training policy, denoted as $\pi_\theta$, which is initialized from a reference policy, $\pi_{init}$. The goal of this optimization is to maximize the rewards obtained from a reward function, $r$. This process seeks to find the optimal parameters $\theta$ by maximizing the expected reward, while also considering the KL divergence between the training policy and the initial policy. The objective function can be expressed as:

$$\max_\theta \mathbb{E}_{\mathbf{y} \sim \pi_\theta(\cdot|\mathbf{x})}[r(\mathbf{x}, \mathbf{y}) - \beta D_{KL}(\pi_\theta(\mathbf{y}|\mathbf{x})||\pi_{init}(\mathbf{y}|\mathbf{x}))] \tag{1}$$

Here, $\mathbf{x}$ is the input, $\mathbf{y}$ the output, $r$ the reward, and $\beta$ is a hyperparameter that balances reward maximization with policy deviation, as measured by the KL divergence $D_{\text{KL}}$.

**Group Relative Policy Optimization (GRPO).** Group Relative Policy Optimization (GRPO) (Shao et al., 2024) bypasses parameterized value models used in traditional methods like Proximal Policy Optimization (PPO). GRPO distinctively calculates policy gradients by weighting trajectory log-likelihoods according to group-based advantages, eliminating the need for a critic model.

In practice, for a given prompt $\mathbf{x}$, GRPO involves sampling $n$ responses (rollouts) $\{\mathbf{y}_1, \mathbf{y}_2, \cdots, \mathbf{y}_n\}$. The reward, $r_i$, associated with each of these $\mathbf{y}_i$ is then used to compute the advantage, $A_i$, for each response $\mathbf{y}_i$.

This advantage is calculated as:

$$A_i = \frac{r_i - \text{mean}(r_1, \ldots, r_n)}{\sqrt{\text{var}(r_1, \ldots, r_n) + \varepsilon}}, \tag{2}$$

where mean($\cdot$) and var($\cdot$) represent the average and variance of the rewards for the $n$ responses, respectively. $\varepsilon > 0$ is a small smoothing constant that ensures the denominator is non-zero.

**Verification and Reward Calculation in RL.** We denote $\mathbf{x}$, $\mathbf{y}_i$, $\mathbf{y}_{ref} \in \mathcal{V}^L$, where $\mathcal{V}$ is the vocabulary space and $L$ is the text length, and $\mathbf{y}_{ref}$ is the ground truth answer to the question $\mathbf{x}$. A verifier is needed to calculate the reward $r_i$ associated with each generated response $\mathbf{y}_i$ for a given question $\mathbf{x}$. Following (Chen et al., 2025a), we model the verifier as an equivalence comparison function:

$$\psi : \mathcal{V}^L \times \mathcal{V}^L \times \mathcal{V}^L \to \{0, 1\}, \qquad \psi(\mathbf{x}, \mathbf{y}_i, \mathbf{y}_{\text{ref}}) = \begin{cases} 1, & \text{if } \mathbf{y}_i \text{ is equivalent to } \mathbf{y}_{\text{ref}} \text{ given } \mathbf{x}, \\ 0, & \text{otherwise.} \end{cases} \tag{3}$$

This function determines if the model's generated response $\mathbf{y}_i$ is equivalent to the ground truth answer $\mathbf{y}_{ref}$. The input prompt $\mathbf{x}$ is optional in this function. The verifier returns 1 if the responses are deemed equivalent and 0 otherwise, providing a binary reward signal for training. The reward $r_i$ is then defined as $r_i = \psi(\mathbf{x}, \mathbf{y}_i, \mathbf{y}_{\text{ref}})$. We note that in practice, we only extract answers within a structured format, e.g., `\boxed{}`, which simplifies verification process.

## 4    Discovering and Analyzing False Negatives from the Wild

In this section, we analyze false negatives in real-world datasets. Specifically, we aim to quantify the prevalence of FNs in answer verification when applying rule-based verifiers.

**Dataset Curation.** We leverage the *Big-Math-RL-Verified dataset* (Apache license 2.0) (Albalak et al., 2025), which comprises over 250,000 diverse, high-quality mathematical problems paired with ground-truth solutions from different sources. Notably, this dataset includes pass rates $p(\mathbf{x})$ for each prompt $\mathbf{x}$ derived from generating 64 responses using LLAMA-3.1-8B, providing an indicator of problem difficulty. To explore false negatives in open-domain settings, we generate $n = 4$ responses per problem using DEEPSEEK-R1-DISTILL-QWEN-7B (Guo et al., 2025), with a temperature of $T = 1$, top-p sampling of $p = 1$, and a context length of $32,768$ tokens. By default, we adopt *Prime Verifier* (Cui et al., 2025), a widely used tool in RL frameworks (e.g., VERL (Sheng et al., 2025)), as the baseline verifier. For our analysis, we retain only the *seemingly incorrect prompt-response pairs* that pass the format check (i.e., has `\boxed{}` in the response) but measured as incorrect by *Prime Verifier*:

$$\mathcal{W} = \big\{(\mathbf{x}, \mathbf{y}_i) : \psi_{\text{prime}}(\mathbf{x}, \mathbf{y}_i, \mathbf{y}_{\text{ref}}) = 0, \mathbf{x} \in \mathcal{X}, i \in \{1, \ldots, n\}\big\}, \tag{4}$$

where $\mathcal{X}$ is the set of all mathematical problems in the dataset.

**False Negative Annotation.** Although *Prime Verifier* accounts for equivalence beyond strict string matching (e.g., LaTeX expression equivalence), it may still misclassify correct answers as incorrect, resulting in false negatives. To systematically investigate these FNs, we employ LLMs to re-evaluate the incorrect responses marked by *Prime Verifier*. To mitigate selection bias and ensure robustness, we select two different LLM annotators: QWEN2.5-72B-INSTRUCT (LLM1) and GROK-3-MINI-HIGH (LLM2), evaluated in non-thinking and thinking modes, respectively. The full prompt can be found in Appendix G.1. We constitute the **false-negative set** by retaining only those prompt-response pairs from $\mathcal{W}$ where both LLMs agree the response is correct:

$$\mathcal{FN} = \big\{(\mathbf{x}, \mathbf{y}_i) \in \mathcal{W} : \psi_{\text{LLM1}}(\mathbf{x}, \mathbf{y}_i, \mathbf{y}_{\text{ref}}) = 1 \ \wedge \ \psi_{\text{LLM2}}(\mathbf{x}, \mathbf{y}_i, \mathbf{y}_{\text{ref}}) = 1\big\}, \tag{5}$$

**Effectiveness of LLM Annotation.** To validate the reliability of our annotation process, we perform a manual review by randomly selecting 200 responses from $\mathcal{FN}$. We observe an accuracy of 99.5%, with only one response incorrectly marked as true due to a missing component in its solution. Additionally, the two

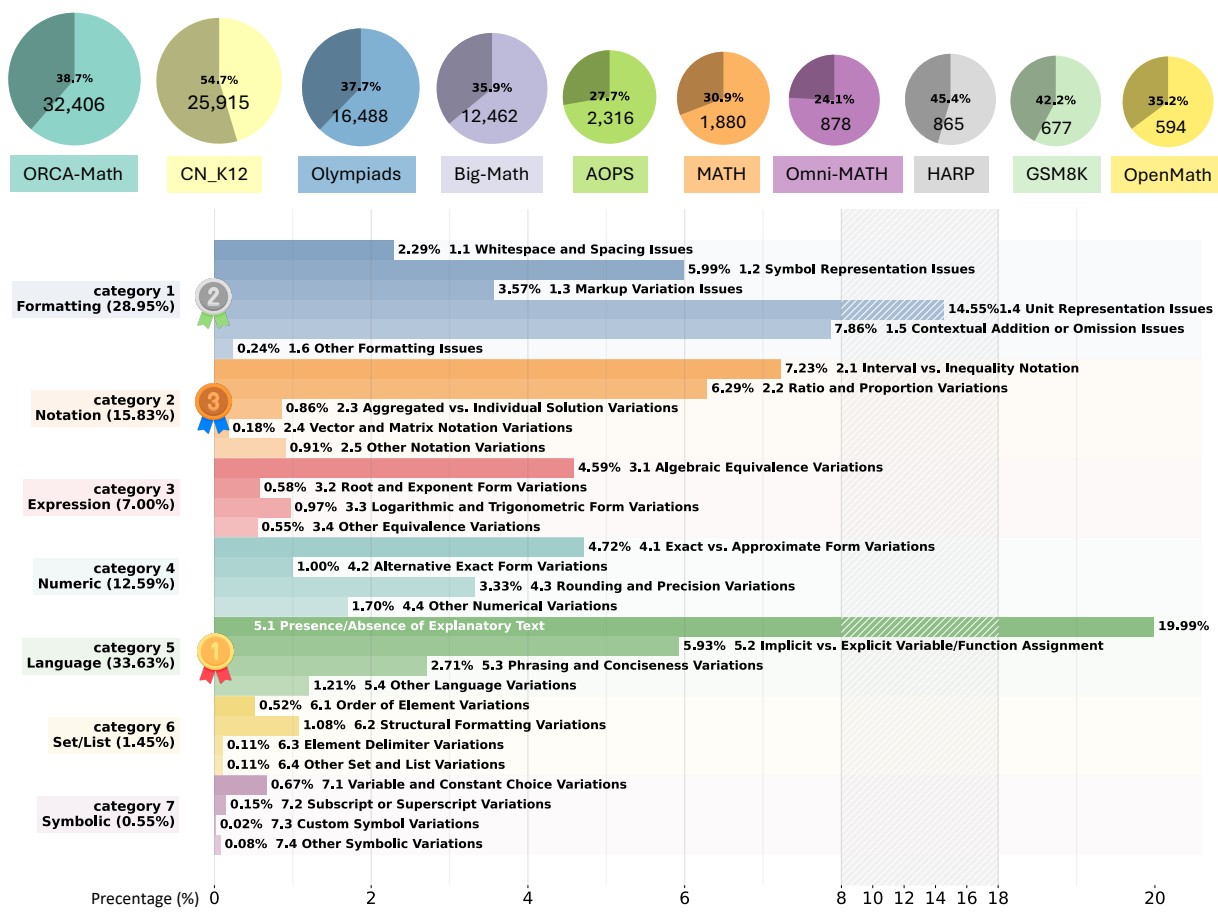

Figure 2: This figure demonstrates false negatives in Big-Math-RL-Verified by source (upper) and category (lower).

LLM verifiers identify three questions with incorrect ground truth answers in the dataset. This indicates that our design can effectively detect false negatives.

**Key Takeaways.** Upon analyzing the false-negative set $\mathcal{FN}$, we have the following key takeaways.

> **Takeaway 1:** High Proportion of False Negatives from the Wild.

Our experiments reveal that, among the 226K prompt-response pairs within seemingly incorrect prompt-response pairs ($\mathcal{W}$), *Prime Verifier* mislabels **87K (38.5%)** correct responses as incorrect. Additionally, among the 95K unique prompts in $\mathcal{W}$, it fails to identify correct responses for **40K (42.1%)** prompts. Figure 2 (upper) shows the false negative ratios across datasets sources, with *CN_K12* exhibiting the highest rate ($> 50\%$).

> **Takeaway 2:** [**Taxonomy of False-Negative Types**] Language differences, formatting inconsistencies, and notation discrepancies are the most prevalent sources of false negatives.

To understand **why** these false negatives occur, we conduct a detailed analysis on $\mathcal{FN}$ and developed a comprehensive taxonomy consisting of seven broad error classes (with 31 finer subcategories), spanning issues from formatting and notation to semantic misunderstandings. We then employ Grok-3-Mini-High to automatically label each prompt exhibiting at least one false negative. The results are demonstrated in Figure 2 (lower). The complete category definitions and annotation prompts are provided in Appendices A and G.2, respectively.

Our analysis reveals that **language differences** constitute the predominant source of false negatives, particularly in cases where either the ground-truth answer or the model-generated response incorporates natural language elements. The second and third most common error sources are **formatting issues** (e.g., missing whitespace or delimiter style) and **notation discrepancies** (e.g., intervals versus inequalities), respectively. The remarkable diversity of these error types underscores the significant challenge faced by rule-based verifiers in attempting to capture all possible variations.

## 5 Analysis of False Negatives and Their Impact on RL Training

### 5.1 Empirical Analysis of FNs during RL

Having examined the distribution of false negatives across datasets in the previous section, we now investigate how these verification errors influence the RL training process.

**RL Training Setups.** We follow Zeng et al. (2025) and perform **zero RL training** on two base models, QWEN2.5-7B and QWEN2.5-MATH-7B, respectively. We follow Ye et al. (2025); Muennighoff et al. (2025) by randomly selecting 5K challenging questions from Big-Math-RL-Verified that satisfy specific difficulty criteria: pass rate $p(\mathbf{x}) \leq 0.2$ for LLAMA-3.1-8B and $p(\mathbf{x}) = 0.25$ for the Deepseek-Distilled models from our curated dataset in Section 4. We perform GRPO (Shao et al., 2024) for 12 epochs with a batch size of 128 and 8 rollouts per sample (i.e., $n = 8$). During training, we employ the default *Prime Verifier* to assign binary rewards based on its verification results. We do not assign additional format rewards during the RL training. Full hyperparameter configurations are detailed in Appendix C.1.

**Methodology.** To systematically investigate false negatives during RL fine-tuning, we adopt the LLM-based false negative annotation outlined in Section 4 and perform an offline evaluation of each rollout generated by the GRPO algorithm. We then compare LLM judgments against the rewards assigned by *Prime Verifier*.

To evaluate how FNs affect GRPO training at each step, we adopt the approach from DAPO (Yu et al., 2025) and define **Prompt Efficiency** $\eta_k$ for a mini-batch of $m$ prompts at training step $k$ as:

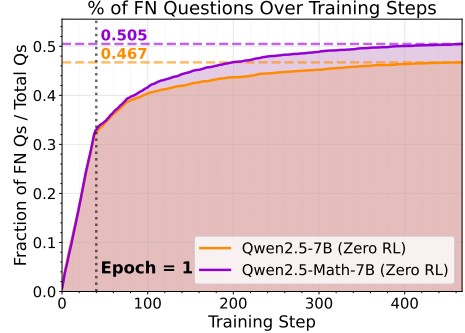

% of FN Questions Over Training Steps

$$\eta_k = P_k(0 < p(\mathbf{x}) < 1) = 1 - P_k(p(\mathbf{x}) = 0) - P_k(p(\mathbf{x}) = 1), \quad (6)$$

where $p(\mathbf{x}) = \frac{1}{n}\sum_{i=1}^{n} r_i$ is the pass rate for a prompt $\mathbf{x}$ with $n$ rollouts, $r_i \in \{0, 1\}$ is the binary reward for the $i$-th rollout, and $P_k$ is the empirical probability over the mini-batch, defined as $P_k(p(\mathbf{x}) = 0) = |\{\mathbf{x} : p(\mathbf{x}) = 0\}|/m$ and $P_k(p(\mathbf{x}) = 1) = |\{\mathbf{x} : p(\mathbf{x}) = 1\}|/m$.

Figure 3: The fraction of unique prompts in the training dataset that encounter at least one false-negative rollout across steps. The x-axis represents the training step, and the y-axis shows the cumulative fraction of prompts affected by false negatives.

Intuitively, prompts for which all rollouts are either correct or incorrect provide no useful gradient signal for RL, whereas partially correct batches are more informative for policy updates. At each training step, we compute Prompt Efficiency using *Prime Verifier*'s reward values and compare these results with the correctness labels derived from our LLM annotations. This comparison enables us to quantify the impact of false negatives on RL training efficiency and overall model performance.

> **Takeaway 3:** High Proportion of False Negative during RL Training.

Figure 3 shows the fraction of unique prompts in the training dataset that experience at least one false-negative rollout across training epochs. The fraction of FN prompts increases steadily after the first epoch, reaching 46.7% for QWEN2.5-7B and 50.5% for QWEN2.5-MATH-7B by the end of training. This trend indicates that false negatives accumulate over time, likely due to the model exploring diverse answer formats that *Prime Verifier* fails to recognize as correct. Moreover, Figure 4 illustrates that the false-negative ratio remains high at every training step, reaching 20% of rollouts on average.

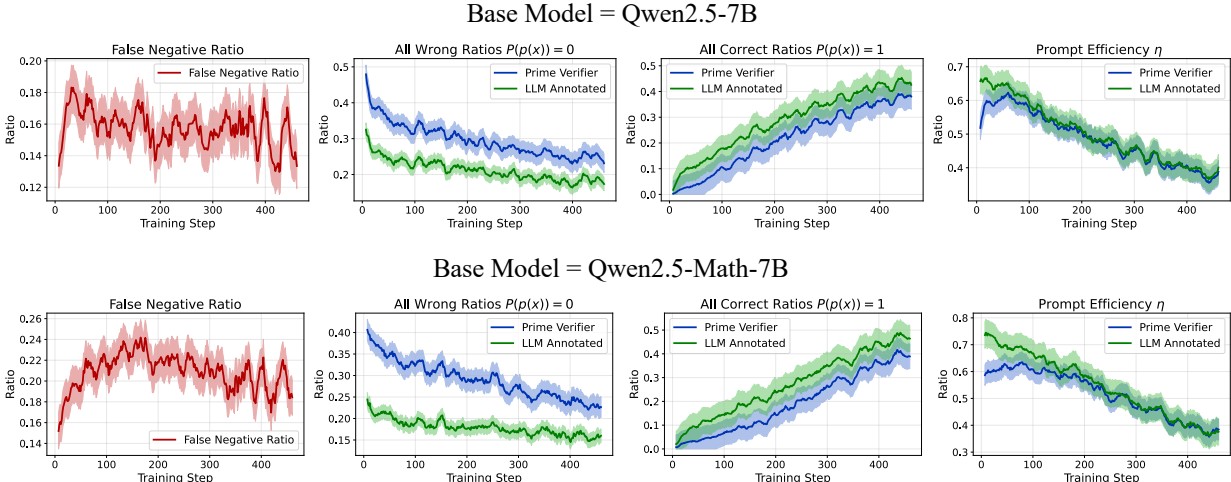

Figure 4: This figure demonstrates the impact of FNs on training efficiency by comparing *Prime Verifier* and LLM annotations. LLM annotations consistently achieve higher prompt efficiency by reducing the all-wrong ratio, particularly in the early stages of training.

> **Takeaway 4:** False Negatives reduce prompt efficiency in early RL training.

Figure 4 illustrates the all-wrong ratio $(P_k(p(\mathbf{x}) = 0))$, all-correct ratio $(P_k(p(\mathbf{x}) = 1))$, and prompt efficiency $\eta_k$ during RL training. We observe that false negatives significantly reduce prompt efficiency $\eta_k$, particularly in the early stages of training. For instance, while *Prime Verifier* marks 50% of prompts as having no correct rollouts, LLM annotations reveal that only 35% lack correct rollouts, indicating a 15% gap. As the all-correct ratio increases with LLM annotations, prompt efficiency based on LLM annotation consistently surpasses that of *Prime Verifier*, driven by a substantial reduction in the all-wrong ratio. We highlight that prompts with low pass rates are more critical for RL training, as they provide informative gradient signals for learning challenging problems (Ye et al., 2025; Muennighoff et al., 2025). Although the gap in prompt efficiency between *Prime Verifier* and LLM annotations narrows in later training stages, *Prime Verifier*'s high false-negative rate in early stages hinders effective learning on challenging prompts.

## 5.2 Theoretical Analysis of Efficiency Degradation Due to False Negatives

In this section, we theoretically analyze the efficiency degradation in GRPO (Shao et al., 2024) caused by false negatives in reward signals. We compare the learnability (defined later) of policies trained with ground truth rewards against those trained with rewards affected by false negatives.

Let $\pi_k^{\mathrm{GT}}(\mathbf{y}_i|\mathbf{x})$ denote the policy optimized at the $k$-th step using ground truth rewards, and let $\pi_k^{\mathrm{FN}}(\mathbf{y}_i|\mathbf{x})$ represent the policy optimized using rewards with false negatives. The success probabilities under these policies for a given prompt $\mathbf{x}$ are defined as:

$$P_k^{\mathrm{GT}} = \mathbb{E}_{\mathbf{y}\sim\pi_k^{\mathrm{GT}}(\cdot|\mathbf{x})}\mathbf{1}_{\{r^{\mathrm{GT}}(\mathbf{y},\mathbf{y}_{\mathrm{ref}})=1\}}, \tag{7}$$

$$P_k^{\mathrm{FN}} = \mathbb{E}_{\mathbf{y}\sim\pi_k^{\mathrm{FN}}(\cdot|\mathbf{x})}\mathbf{1}_{\{r^{\mathrm{FN}}(\mathbf{y},\mathbf{y}_{\mathrm{ref}})=1\}}, \tag{8}$$

where $\mathbf{1}_{\{\cdot\}}$ is the indicator function, $r^{\mathrm{GT}}(\mathbf{y}, \mathbf{y}_{\mathrm{ref}})$ is the ground truth reward function, and $r^{\mathrm{FN}}(\mathbf{y}, \mathbf{y}_{\mathrm{ref}})$ is the reward function affected by false negatives.

Given the definition of false negatives, where a correct response may be incorrectly marked as incorrect, we have the following lemma.

**Lemma 1.** $P_k^{GT} > P_k^{FN}$ *for all* $k$.

Our theoretical framework relies on the following two assumptions:

**Assumption 1.** $P_k^{\mathrm{GT}}$ increases with $k$.

This assumption posits that the GRPO is fundamentally sound, ensuring that the success probability (i.e., average reward scores) improves over iterations when trained with ground truth rewards.

**Assumption 2.** $P_k^{\mathrm{GT}} < 2P_{k-1}^{\mathrm{GT}}$ for all $k$.

This assumes that the average reward scores will not grow exponentially during training, which is consistent with the practical improvement of reward scores in reinforcement learning policy updates.

Following Bae et al. (2025), we define step-wise learnability as the reverse KL divergence between policies at consecutive optimization steps, denoted by $D_k$. For a policy trained with ground truth rewards and rewards containing false negatives, the step-wise learnability is:

$$D_{k,\mathrm{GT}} = D_{\mathrm{KL}}\big(\pi_{k-1}^{\mathrm{GT}}(\mathbf{y}|\mathbf{x}) \,\|\, \pi_k^{\mathrm{GT}}(\mathbf{y}|\mathbf{x})\big), \tag{9}$$

$$D_{k,\mathrm{FN}} = D_{\mathrm{KL}}\big(\pi_{k-1}^{\mathrm{FN}}(\mathbf{y}|\mathbf{x}) \,\|\, \pi_k^{\mathrm{FN}}(\mathbf{y}|\mathbf{x})\big). \tag{10}$$

These metrics quantify improvement in policy distribution between consecutive steps. Specifically, the reverse KL divergence measures the distance between the previous policy $\pi_{k-1}$ and the updated policy $\pi_k$, where a larger $D_k$ indicates greater policy improvement and thus better learnability.

Our main theoretical result is encapsulated in the following theorem:

**Theorem 1.** *Let $\delta_k = D_{k,GT} - D_{k,FN}$ denote the step-wise learnability gap at training step $k$. Under Lemma 1 and Assumption 1, $\delta_k > 0$ for all $k$.*

The proof is provided in Appendix B. This theorem shows that policies trained with ground truth rewards have greater step-wise learnability than those with false negatives, highlighting the importance of accurate reward signals in RL, as false negatives impede convergence.

## 6 Improve RL by Detecting False Negatives with TinyV

Our experimental and theoretical analysis demonstrate that false negatives are a pervasive issue in RL training, severely impacting training efficiency. While LLM-based annotators like QWEN2.5-72B-INSTRUCT and GROK-3-MINI-HIGH can effectively identify false negatives, this approach is computationally expensive, economically infeasible, and introduces delays due to the high resource demands of large-scale LLMs. To address these limitations, we propose TINYV, a lightweight LLM-based verifier that augments existing rule-based methods like *Prime Verifier*, which dynamically identifies potential FNs and recovers valid responses, enabling more accurate reward estimates while maintaining computational efficiency.

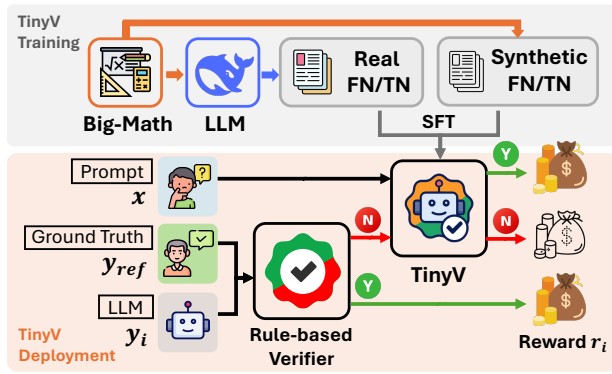

Figure 5: This figure demonstrates the curation and deployment of TINYV.

### 6.1 Curation of TinyV

In this subsection, we outline the process for creating TINYV, focusing on dataset curation, model training, and deployment setup.

**Dataset Curation.** To develop a reliable verifier capable of handling diverse scenarios, we curate a hybrid dataset comprising both **real** and **synthetic** examples of false negatives and true negatives. The real false negative and true negative data are sourced from Section 4, where the correctness of the responses were annotated by LLMs. To ensure broader coverage and robustness, we augment this dataset with synthetically

generated false negatives. Specifically, we prompt QWEN2.5-72B-INSTRUCT to generate potential false negative cases for a given question by introducing variations such as LaTeX formatting differences, numerical approximations, or alternative mathematical expressions that preserve semantic equivalence. These generated candidates are then re-annotated by LLMs to confirm they are false negatives. The detailed data curation process, including the prompts used, is provided in Appendix C.2. In total, we collect 638,000 instances, each consisting of a prompt, ground truth, model answer, and LLM-annotated correctness label. This hybrid approach ensures that TINYV can generalize across a wide range of false negative patterns.

**Model Training.** We perform supervised fine-tuning on *Qwen2.5-1.5B-Instruct*, a compact model selected to balance performance and computational efficiency. The training employs a binary classification setup, where the model predicts a label of "True" for a response that is correct (i.e., a false negative when flagged as incorrect by *Prime Verifier*) and "False" otherwise. The inputs are model's answer, the ground truth, and the problem context. To ensure a balanced dataset and mitigate bias, we sample 159,000 instances, equally distributed between "True" and "False" labels. The training template, hyperparameters, and configurations are detailed in Appendix C.3. Appendix C.4 further compares TINYV against its teacher LLMs as well as other strong baselines, with a particular focus on FN rates. Additionally, we experiment with training TINYV-THINK, a variant that performs intermediate analysis before predicting the final label. However, this approach introduces significant delays due to longer generation time, making it less practical for RL. Consequently, we adopt TINYV for our main experiments. A detailed comparison between TINYV and TINYV-THINK is provided in Appendix E.1.

**TinyV Deployment.** To maximize efficiency and align with Theorem 1, we integrate TINYV in an **add-on** mode alongside *Prime Verifier*, as shown in Figure 5. In this configuration, TINYV is queried only when *Prime Verifier* returns a negative result (i.e., flags a response as incorrect). TINYV then re-evaluates the response to determine if it is a false negative, thus avoiding unnecessary computations for responses already deemed correct. This hierarchical setup ensures that TINYV complements *Prime Verifier* by focusing computational resources on challenging cases, thereby enhancing the accuracy of reward signals in RL training while minimizing overhead.

## 6.2 HardVerify-Math Benchmark

While existing mathematical benchmarks have advanced the evaluation of LLMs in reasoning tasks, they often consist of questions with easily verifiable answers, such as simple numerical solutions. This limitation highlights the need for a new benchmark that focuses on challenging verification scenarios prone to false negatives. To address this, we curate the *HardVerify-Math Bench*, a benchmark comprising 250 hard-to-verify answers spanning all categories and the taxonomy discussed in Section 4. Specifically, we manually select 115 questions from Olympiad benchmark and 10 questions from the MATH test sets that are prone to false negative cases due to their complexity in answer format. Additionally, we include 125 questions from the *Big-Math* dataset, chosen based on a Llama-3.1-8B pass rate of less than 0.05 and identified as challenging to verify by human experts. A detailed introduction to this benchmark including its distribution and examples is in Appendix D.

## 6.3 Experimental Setups

**Models and Datasets.** In our main experiments, we use five models including *Qwen3-1.7B-base*, *Qwen2.5-3B*, *Qwen2.5-7B*, *Qwen2.5-Math-7B*, and *DeepSeek-Math-7B-Instruct*, and perform zero-RL training using GRPO (Shao et al., 2024). For training, we sample 5,000 questions from the *Big-Math* dataset that exhibit false negative cases, with pass rates satisfying $0.05 < p(\mathbf{x}) \leq 0.2$ for LLAMA-3.1-8B and $p(\mathbf{x}) \leq 0.25$ for DeepSeek-Distilled models. These criteria ensure sufficient challenge while avoiding overlap with our *HardVerify-Math* benchmark. We employ TINYV and *Prime Verifier* to assign rewards. For comparative analysis, we randomly sample 5,000 questions from *DeepScaleR* (Luo et al., 2025), which contains questions with easily verifiable answers (e.g., plain numerical values or simple formats evaluable using the SYMPY library), and use *Prime Verifier* for evaluation due to its simplicity in answer verification.

In our ablation analysis, we extend beyond standard setups to include several more dimensions:

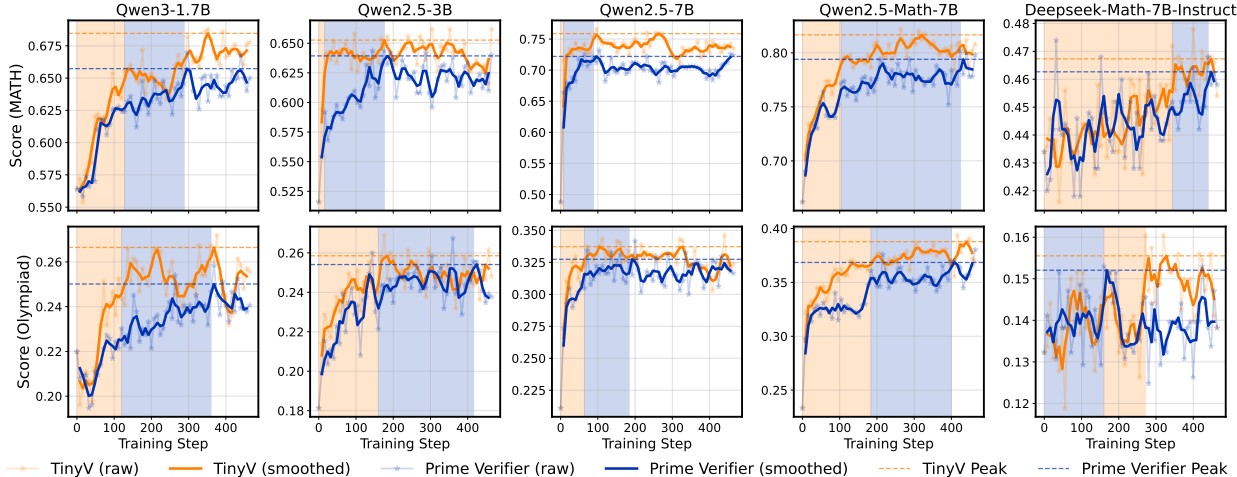

Figure 6: Performance trends of five models on the MATH and Olympiad benchmark, comparing TINYV with *Prime Verifier*. The darker lines are smoothed using a sliding window whose size is 5% of the total training steps, and the dashed lines indicate the peak performance (smoothed score) achieved during training. The colored vertical shaded areas illustrate the training steps required for TINYV to reach the *Prime Verifier*'s peak score (Orange), and for the Prime Verifier to reach its own peak score (Blue). We observe that the model trained with TINYV converges faster and has better peak and final model performance. **On average, TinyV reaches the peak performance of *Prime Verifier* using only 49.7% RL steps, indicating its higher efficiency.**

- **Baseline Verifiers.** We incorporate additional state-of-the-art baseline verifiers, including Math, MathRuler (hiyouga, 2025), and Math-Verify (Hugging Face, 2025), to provide a robust comparison with TINYV.

- **Out-of-Domain (OOD) Generalization.** We evaluate TINYV robustness by testing its performance on Out-of-Domain datasets, i.e., DeepMath (He et al., 2025) and Natural Reasoning (Yuan et al., 2025a).

- **RL Algorithm Comparison.** We apply Proximal Policy Optimization (PPO) (Schulman et al., 2017) to test the effectiveness of the proposed TINYV.

**Benchmarks and Evaluation Setups.** We assess the performance of trained models on MATH500 (Hendrycks et al., 2021), AMC (2023 and 2024), Olympiad Bench (He et al., 2024a), and *HardVerify-Math*. All experiments employ greedy decoding to ensure deterministic and reproducible results. For MATH500, AMC, and the Olympiad Bench, we adopt the standard practice of using Prime Verify for answer verification. For the more challenging HardVerify-Math, we instead employ LLM-based evaluations to assess performance. We further use GPQA-Diamond (Rein et al., 2024) to evaluate the performance of TINYV on OOD tasks. More experimental setups can be found in Appendix C.1.

## 6.4 Experimental Results

In this subsection, we present a summary of our experimental results, highlighting the improvements achieved by TINYV in RL training efficiency and model performance across various benchmarks.

> **Takeaway 5:** TINYV enhances RL training efficiency and final model performance. On average, TINYV reaches the peak performance of *Prime Verifier* using **less than 50% RL training steps**.

As shown in Figure 6 and Table 1, TINYV significantly enhances the efficiency of RL training compared to *Prime Verifier*, achieving faster convergence and reducing the training steps required to reach *Prime Verifier*'s peak performance. Furthermore, the peak and final model performance of TINYV consistently

Table 1: Final performance comparison across different base models with Prime Verifier and TINYV-based setups. The models are trained using **GRPO** algorithm. Values represent accuracy percentages, with the best performance per base model highlighted in **bold**.

| Base Model | Experiment Setup | HardVerify-Math | MATH | AMC | Olympiad | Average |
|---|---|---|---|---|---|---|
| *Qwen3-1.7B* | Base | 37.8% | 56.4% | 32.5% | 22.0% | 37.2% |
| | + Prime Verifier | 45.3% | **65.0%** | 31.3% | 24.1% | 41.4% |
| | + TINYV | **51.0%** | **65.0%** | **33.7%** | **25.6%** | **44.5%** |
| *Qwen2.5-3B* | Base | 33.7% | 51.6% | 30.1% | 18.1% | 33.4% |
| | Prime Verifier | 47.0% | 64.0% | 34.9% | 23.7% | 43.0% |
| | + TINYV | **49.8%** | **66.2%** | **37.3%** | **24.8%** | **43.9%** |
| *Qwen2.5-7B* | Base | 41.0% | 46.8% | 25.3% | 22.1% | 33.8% |
| | + Prime Verifier | 58.6% | 72.4% | **44.6%** | 31.7% | 51.8% |
| | + TINYV | **68.7%** | **73.4%** | 43.4% | **32.4%** | **54.5%** |
| *Qwen2.5-Math-7B* | Base | 50.6% | 67.8% | 48.1% | 23.6% | 47.5% |
| | + Prime Verifier | 62.7% | 79.8% | 48.2% | **38.0%** | 57.2% |
| | + TINYV | **69.1%** | **80.8%** | **53.0%** | 37.0% | **60.0%** |
| *Deepseek-Math-7B-Instruct* | Base | 26.1% | 43.4% | 17.1% | 13.2% | 25.0% |
| | + Prime Verifier | 30.1% | **45.8%** | 12.0% | **13.8%** | 25.4% |
| | + TINYV | **32.1%** | 45.4% | **16.9%** | **13.8%** | **27.1%** |

outperforms that of *Prime Verifier* across almost all training steps, with a performance gap of up to 10% in some benchmarks. We attribute this improvement to TINYV 's ability to provide more accurate reward signals, enabling the model to learn effectively from challenging questions where *Prime Verifier* often fails to detect correct responses.

> **Takeaway 6:** TINYV improves performance on *HardVerify-Math* compared to baselines.

As shown in Figure 7, TINYV trained on the *Big-Math* dataset outperforms the baseline using *DeepScaleR* on the *HardVerify-Math* benchmark. Notably, the performance of *DeepScaleR* on HardVerify-Math fluctuates, likely due to its focus on easily verifiable questions that do not generalize well to hard-to-verify scenarios. In contrast, both TINYV and *Prime Verifier* with *Big-Math* show consistent improvement, with TINYV achieving a final accuracy of 68.68% compared to *Prime Verifier*'s 58.64% with *Qwen2.5-7B* as the base model. We attribute this to *DeepScaleR*'s limitation in training on questions with simple, clean answers, which leaves the model underprepared for the complex, false negative-prone questions in *HardVerify-Math*. Interestingly, this performance advantage of TINYV extends to other benchmarks like MATH500 and Olympiad Bench, where some solutions are similarly challenging to verify due to their complexity (e.g., symbolic expressions or sets). This suggests a gap in current training datasets that fail to address hard-to-verify scenarios, opening avenues for future research into developing more diverse datasets and adaptive verification methods that can better handle such challenges.

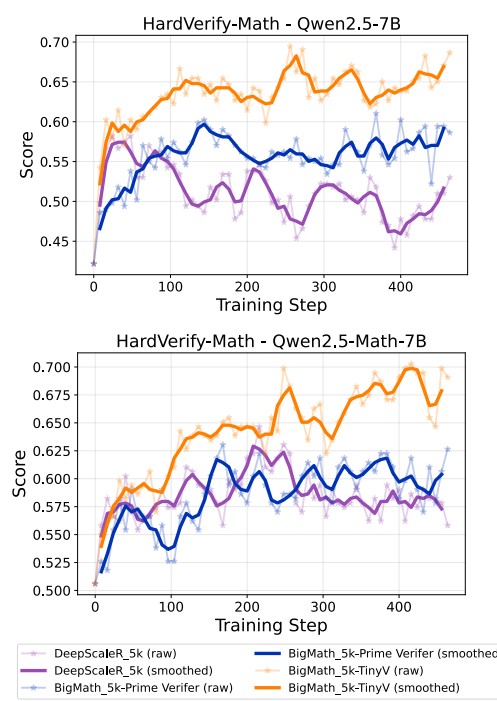

Figure 7: This figure compares performance of *HardVerify-Math* between *Big-Math* (hard to verify) and *DeepScaleR* (easy to verify) datasets.

> **Takeaway 7:** TINYV generalizes across RL algorithms.

While our main experiments use GRPO, false negatives influence reward assignment in a way that is independent of the underlying RL algorithm. To assess whether the benefits of

Table 2: Final performance comparison under **PPO** across different base models with Prime Verifier and TINYV-based setups. Values represent accuracy percentages, with the best performance per base model highlighted in **bold**.

| Base Model | Experiment Setup | HardVerify-Math | MATH | AMC | Olympiad | Average |
|---|---|---|---|---|---|---|
| *Qwen2.5-3B* | Base | 33.7% | 51.6% | 30.1% | 18.1% | 33.4% |
| | + Prime Verifier | 39.8% | 59.2% | 33.7% | 22.6% | 38.8% |
| | + TINYV | **48.6%** | **65.4%** | 32.5% | **25.1%** | **42.9%** |
| *Qwen2.5-Math-7B* | Base | 50.6% | 67.8% | 48.1% | 23.6% | 47.5% |
| | + Prime Verifier | 57.4% | 78.2% | 54.2% | 34.0% | 56.0% |
| | + TINYV | **66.7%** | **78.6%** | **57.8%** | **37.6%** | **60.2%** |
| *Qwen2.5-7B* | Base | 41.0% | 46.8% | 25.3% | 22.1% | 33.8% |
| | + Prime Verifier | 56.6% | 70.8% | **44.5%** | 31.2% | 50.8% |
| | + TINYV | **59.8%** | **73.6%** | 40.4% | **31.6%** | **51.4%** |

TINYV are specific to GRPO, we perform an ablation using Proximal Policy Optimization (PPO) (Schulman et al., 2017) across three base models.

For each base model, we compare (i) the pretrained model, (ii) PPO fine-tuning with the rule-based *Prime Verifier*, and (iii) PPO fine-tuning with TINYV. Across all model families, TINYV consistently achieves higher average performance than Prime Verifier. This indicates that TINYV's false-negative recovery improves reward quality in a manner that transfers beyond GRPO, and generalizes across both RL algorithms and model scales.

### 6.5 Additional Experimental Results

We compare the performance of different verifiers, including TINYV, TINYV-THINK, Math, MathRuler (hiyouga, 2025), and Math-Verify (Hugging Face, 2025), in Appendix E.1. We further evaluate TINYV against strong LLM-judge baselines (GPT-4o-mini, DeepSeek-V3.2, and MiMo-V2-Flash) in Appendix E.2. These experiments show that TINYV achieves competitive performance with much larger commercial and open-weight judges while incurring significantly lower computational cost. We also compare training costs with and without TINYV in Appendix E.3. Our analysis shows that TINYV incurs only an average 6% overhead per training step, confirming its lightweight design. Appendix E.4 and Appendix E.5 study generalization to out-of-distribution mathematical data and non-mathematical domains, respectively. Appendix E.6 analyzes the effect of false positives on RL training.

## 7 Conclusion and Future Work

In this work, we study false negatives (FNs) in reinforcement learning with verifiable rewards, focusing on three core questions regarding their **prevalence**, **impact**, and **mitigation**. We show that FNs are widespread in RL training and significantly affect prompt efficiency from both empirical and theoretical perspectives. We further demonstrate that the proposed TINYV improves reward accuracy while remaining computationally lightweight, leading to faster convergence and stronger final performance than rule-based verifiers.

Future work could explore false negatives in broader RL domains, such as theorem proving (Xin et al., 2024), medical applications (Lai et al., 2025), software engineering development (Wei et al., 2025), and robotics (Boyle et al., 2025), to further enhance the robustness and generalizability of RL training across diverse reasoning and decision-making tasks.

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

# A    Detailed False Negative Categories

In this section, we present a comprehensive taxonomy of false negatives identified in answer verification for mathematical reasoning tasks, based on our analysis on the *Big-Math-RL-Verified dataset*. These categories highlight the diverse reasons why rule-based verifiers, such as *Prime Verifier*, may incorrectly mark a model's response as wrong despite it being mathematically correct. Each category is divided into subcategories, with descriptions and illustrative examples to demonstrate the variations leading to false negatives.

## A.1    Formatting and Syntax Differences

This category captures differences in formatting and syntax that do not alter the mathematical meaning of the answer.

- **Formatting → Whitespace and Spacing Issues**

  - *Description:* Variations in spaces around operators, within expressions, or between elements.
  - *Example:*
    ```
    ground truth answer: f(x) = 2 x
    model answer: f(x)=2x
    ```

- **Formatting → Symbol Representation Issues**

  - *Description:* Differences in symbol notation, including Unicode vs. command-based symbols, delimiter styles, or minor symbol variations (e.g., degree symbols, infinity notation).
  - *Example:*
    ```
    ground truth answer: (−∞, -3) ∪ (3, +∞)
    model answer: (−∞, -3) ∪ (3, ∞)
    ```

- **Formatting → Markup Variation Issues**

  - *Description:* Differences in syntax for equivalent rendering, such as LaTeX command choices or delimiter sizing.
  - *Example:*
    ```
    ground truth answer: \frac{32}{9}
    model answer: \dfrac{32}{9}
    ```

- **Formatting → Unit Representation Issues**

  - *Description:* Differences in the inclusion, omission, or representation of units (e.g., missing units, abbreviated vs. full unit names).
  - *Example:*
    ```
    ground truth answer: 18.8^\circ
    model answer: 18.8
    ```

- **Formatting → Contextual Addition or Omission Issues**

  - *Description:* Missing or extra prefixes (e.g., "x=") or explanatory text not affecting the core answer, excluding units.
  - *Example:*
    ```
    ground truth answer: N=n
    model answer: n
    ```

- **Formatting → Other Formatting Issues**

  - *Description:* Miscellaneous formatting differences, such as newline characters or non-alphanumeric separators.
  - *Example:*
    ```
    ground truth answer: 60^\text{circ} 42'
    model answer: 60^\circ 42'
    ```

## A.2 Mathematical Notation Variations

This category includes differences in standard mathematical conventions for expressing the same concept.

- **Notation → Interval vs. Inequality Notation**

    - *Description:* Representing ranges as intervals or inequalities.
    - *Example:*
      ground truth answer: $(-\infty, \text{-5})$
      model answer: k < -5

- **Notation → Ratio and Proportion Variations**

    - *Description:* Different ways of expressing ratios or proportions (e.g., colon, fraction, or single value).
    - *Example:*
      ground truth answer: 2:1
      model answer: 2/1

- **Notation → Aggregated vs. Individual Solution Variations**

    - *Description:* Using symbols like $\pm$ or listing solutions separately.
    - *Example:*
      ground truth answer: 1 ± \sqrt{19}
      model answer: 1 + \sqrt{19}, 1 - \sqrt{19}

- **Notation → Vector and Matrix Notation Variations**

    - *Description:* Variations in displaying vectors or matrices.
    - *Example:*
      ground truth answer: \begin{pmatrix} -7 \\16 \\5 \end{pmatrix}
      model answer: (-7,16,5)

- **Notation → Other Notation Variations**

    - *Description:* Variations due to regional conventions (e.g., decimal points vs. commas) or other notation differences.
    - *Example:*
      ground truth answer: 3.14
      model answer: 3,14

## A.3 Mathematical Expression Equivalencies

This category covers expressions that differ in form but are mathematically equivalent.

- **Expression → Algebraic Equivalence Variations**

    - *Description:* Different but equivalent algebraic forms, including term ordering, factoring, or simplification.
    - *Example:*
      ground truth answer: \frac{1-p^{2}}{3}
      model answer: \frac{-p^2+1}{3}

- **Expression → Root and Exponent Form Variations**

    - *Description:* Using roots, fractional exponents, or simplified exponents differently.
    - *Example:*
      ground truth answer: 2^{-2 / 3}
      model answer: \frac{1}{\sqrt[3]{4}}

- **Expression → Logarithmic and Trigonometric Form Variations**

  - *Description:* Equivalent forms using logarithmic or trigonometric identities.
  - *Example:*
    ```
    ground truth answer: \frac{\log 2}{\log 2-\log 3}
    model answer: -\frac{\ln 2}{\ln 3-\ln 2}
    ```

- **Expression → Other Equivalence Variations**

  - *Description:* Equivalencies in combinatorial quantities, complex numbers, or other mathematical structures.
  - *Example:*
    ```
    ground truth answer: \frac{3 m}{2}-1
    model answer: \dfrac{3m - 2}{2}
    ```

## A.4 Numerical Representation Differences

This category addresses variations in how numerical values are presented.

- **Numeric → Exact vs. Approximate Form Variations**

  - *Description:* Exact (fraction, symbolic) vs. decimal or percentage approximations.
  - *Example:*
    ```
    ground truth answer: \frac{600}{7}
    model answer: 85.71
    ```

- **Numeric → Alternative Exact Form Variations**

  - *Description:* Different exact representations, such as scientific notation or evaluated powers.
  - *Example:*
    ```
    ground truth answer: 10^{3}
    model answer: 1000
    ```

- **Numeric → Rounding and Precision Variations**

  - *Description:* Approximations with different decimal places or rounding rules.
  - *Example:*
    ```
    ground truth answer: 1.27\%
    model answer: 1.3\%
    ```

- **Numeric → Other Numerical Variations**

  - *Description:* Other numerical format differences, such as mixed vs. improper fractions.
  - *Example:*
    ```
    ground truth answer: 6\frac{1}{64}
    model answer: 6.015625
    ```

## A.5 Language and Contextual Variations

This category captures differences in natural language or implied context.

- **Language → Presence/Absence of Explanatory Text**

  - *Description:* Model output or ground truth includes additional descriptive text, or vice versa.
  - *Example:*
    ```
    ground truth answer: 10,11,12,13,14,-2,-1,0,1,2
    model answer: Sequence 1:  -2, -1, 0, 1, 2 and Sequence 2:  10, 11, 12, 13, 14
    ```

- **Language → Implicit vs. Explicit Variable/Function Assignment**

  - *Description:* One output explicitly assigns values to variables or defines a function while the other lists values or the expression directly.
  - *Example:*
    ```
    ground truth answer: 16,3,1,1
    model answer: w=16, d=3, a=1, b=1
    ```

- **Language → Phrasing and Conciseness Variations**

  - *Description:* Differences in wording, synonyms, or level of detail.
  - *Example:*
    ```
    ground truth answer: \text{Any odd number of participants}
    model answer: odd
    ```

- **Language → Other Language Variations**

  - *Description:* Minor differences in separators (e.g., "and" vs. comma) or answer structure.
  - *Example:*
    ```
    ground truth answer: 1,3
    model answer: 1 \text{ and } 3
    ```

## A.6 Set and List Differences

This category includes variations in presenting collections of results, assuming correctness.

- **Set/List → Order of Element Variations**

  - *Description:* Different sequencing of elements in sets or lists where order is not mathematically significant.
  - *Example:*
    ```
    ground truth answer: (6,3),(9,3),(9,5),(54,5)
    model answer: (9,3),(6,3),(54,5),(9,5)
    ```

- **Set/List → Structural Formatting Variations**

  - *Description:* Variations in tuple, set, or list formatting, including use of braces.
  - *Example:*
    ```
    ground truth answer: (1,2), (3,4)
    model answer: {(1,2), (3,4)}
    ```

- **Set/List → Element Delimiter Variations**

  - *Description:* Differences in delimiters used to separate elements (e.g., commas vs. semicolons).
  - *Example:*
    ```
    ground truth answer: (1,2,3)
    model answer: (1;2;3)
    ```

- **Set/List → Other Set and List Variations**

  - *Description:* Other differences in set or list presentation, such as redundant parentheses.
  - *Example:*
    ```
    ground truth answer: (1,2)
    model answer: ((1,2))
    ```

### A.7 Symbolic Representation Variations

This category addresses differences in variable or constant symbols.

- **Symbolic → Variable and Constant Choice Variations**

  - *Description:* Different letters or cases for arbitrary constants or parameters.
  - *Example:*
    ```
    ground truth answer: ...+\pi k, ...
    model answer: ...+n \pi, ...
    ```

- **Symbolic → Subscript or Superscript Variations**

  - *Description:* Differences in subscript or superscript notation for variables or constants.
  - *Example:*
    ```
    ground truth answer: x_1, x_2
    model answer: x^1, x^2
    ```

- **Symbolic → Custom Symbol Variations**

  - *Description:* Use of unconventional or user-defined symbols for variables or constants.
  - *Example:*
    ```
    ground truth answer: α, β
    model answer: a, b
    ```

- **Symbolic → Other Symbolic Variations**

  - *Description:* Other differences in symbolic representation, such as case sensitivity.
  - *Example:*
    ```
    ground truth answer: P(x)
    model answer: p(x)
    ```

## B Proof of Theorem 1

In this section, we provide a detailed proof of Theorem 1, which states that policies trained with ground truth rewards have greater step-wise learnability than those with false negatives. We first derive the closed-form expression of the step-wise learnability in Section B.1, and then prove the positivity of the step-wise learnability gap in Sections B.2 and B.3.

### B.1 Reverse KL for GRPO Updates

We begin with the GRPO objective

$$\max_{\theta} \; \mathbb{E}_{\mathbf{y} \sim \pi_\theta(\cdot|\mathbf{x})}\big[r(\mathbf{x}, \mathbf{y})\big] - \beta \, D_{\mathrm{KL}}\big(\pi_\theta(\mathbf{y} \mid \mathbf{x}) \,\|\, \pi_{\mathrm{init}}(\mathbf{y} \mid \mathbf{x})\big),$$

and transform this optimization into a step-wise recursion. Throughout, we denote:

- $\mathbf{x}$: input prompt

- $\mathbf{y}$: output token/sequence

- $r(\mathbf{x}, \mathbf{y}) \in \{0, 1\}$: binary reward

- $p_k(\mathbf{x}) = \mathbb{E}_{\mathbf{y} \sim \pi_k(\cdot|\mathbf{x})}\big[\mathbf{1}_{\{r(\mathbf{x},\mathbf{y})=1\}}\big]$: success probability of policy $\pi_k$ for prompt $\mathbf{x}$

- $p_{\mathrm{ref}}(\mathbf{x}) = \mathbb{E}_{\mathbf{y} \sim \pi_{\mathrm{ref}}(\cdot|\mathbf{x})}\big[\mathbf{1}_{\{r(\mathbf{x},\mathbf{y})=1\}}\big]$: success probability of reference policy for prompt $\mathbf{x}$

**Lemma 2** (GRPO Policy Dynamics Mroueh (2025)). *For $k \geq 1$, the optimal GRPO iterate satisfies*

$$\pi_k(\mathbf{y} \mid \mathbf{x}) = \frac{1}{Z_{k-1}(\mathbf{x})} \, \pi_{\text{ref}}(\mathbf{y} \mid \mathbf{x}) \, \exp\!\Big(\tfrac{1}{\beta}\big[\omega_\varepsilon^+\big(p_{k-1}(\mathbf{x})\big)\,\mathbf{1}_{\{r(\mathbf{x},\mathbf{y})=1\}} - \omega_\varepsilon^-\big(p_{k-1}(\mathbf{x})\big)\,\mathbf{1}_{\{r(\mathbf{x},\mathbf{y})=0\}}\big]\Big)$$

*with weights*

$$\omega_\varepsilon^+(p) = \frac{1-p}{\sqrt{p(1-p)}+\varepsilon}, \qquad \omega_\varepsilon^-(p) = \frac{p}{\sqrt{p(1-p)}+\varepsilon},$$

*and normalizing constant*

$$Z_{k-1}(\mathbf{x}) = p_{\text{ref}}(\mathbf{x})\, e^{\frac{1}{\beta}\omega_\varepsilon^+\big(p_{k-1}(\mathbf{x})\big)} + \big(1 - p_{\text{ref}}(\mathbf{x})\big)\, e^{-\frac{1}{\beta}\omega_\varepsilon^-\big(p_{k-1}(\mathbf{x})\big)}.$$

*Proof.* See Mroueh (2025) for the proof. $\qquad\square$

Building on Lemma 2, we now derive the reverse Kullback–Leibler (KL) divergence between two consecutive GRPO iterates.

**Lemma 3** (Reverse KL for GRPO Updates). *Given the GRPO policy updates from Lemma 2, the reverse KL divergence satisfies*

$$D_{\text{KL}}\big(\pi_{k-1}(\cdot \mid \mathbf{x}) \,\|\, \pi_k(\cdot \mid \mathbf{x})\big) = \frac{1}{\beta}\Big[\omega_\varepsilon^+\big(p_{k-2}(\mathbf{x})\big)\,p_{k-1}(\mathbf{x}) - \omega_\varepsilon^-\big(p_{k-2}(\mathbf{x})\big)\,\big(1 - p_{k-1}(\mathbf{x})\big)\Big] - \log\frac{Z_{k-2}(\mathbf{x})}{Z_{k-1}(\mathbf{x})}$$

*Proof.* By definition, the reverse KL divergence between $\pi_{k-1}$ and $\pi_k$ is:

$$D_{\text{KL}}\big(\pi_{k-1}(\cdot \mid \mathbf{x}) \,\|\, \pi_k(\cdot \mid \mathbf{x})\big) = \sum_{\mathbf{y}} \pi_{k-1}(\mathbf{y} \mid \mathbf{x}) \log \frac{\pi_{k-1}(\mathbf{y} \mid \mathbf{x})}{\pi_k(\mathbf{y} \mid \mathbf{x})}.$$

Using the GRPO update rule from Lemma 2 for both policies, we can express $\pi_{k-1}$ and $\pi_k$ as:

$$\pi_{k-1}(\mathbf{y} \mid \mathbf{x}) = \frac{1}{Z_{k-2}(\mathbf{x})} \, \pi_{\text{ref}}(\mathbf{y} \mid \mathbf{x}) \, \exp\!\Big(\tfrac{1}{\beta}\big[\omega_\varepsilon^+\big(p_{k-2}(\mathbf{x})\big)\,\mathbf{1}_{\{r(\mathbf{x},\mathbf{y})=1\}} - \omega_\varepsilon^-\big(p_{k-2}(\mathbf{x})\big)\,\mathbf{1}_{\{r(\mathbf{x},\mathbf{y})=0\}}\big]\Big)$$

and similarly for $\pi_k(\mathbf{y} \mid \mathbf{x})$. Taking the log-ratio and simplifying the result, we get:

$$\log \frac{\pi_{k-1}(\mathbf{y} \mid \mathbf{x})}{\pi_k(\mathbf{y} \mid \mathbf{x})} = \log \frac{Z_{k-1}(\mathbf{x})}{Z_{k-2}(\mathbf{x})} + \tfrac{1}{\beta}\big[\Delta_k^+(\mathbf{x})\,\mathbf{1}_{\{r(\mathbf{x},\mathbf{y})=1\}} - \Delta_k^-(\mathbf{x})\,\mathbf{1}_{\{r(\mathbf{x},\mathbf{y})=0\}}\big] \tag{11}$$

where we denote:

$$\Delta_k^+(\mathbf{x}) = \omega_\varepsilon^+\big(p_{k-2}(\mathbf{x})\big) - \omega_\varepsilon^+\big(p_{k-1}(\mathbf{x})\big), \quad \Delta_k^-(\mathbf{x}) = \omega_\varepsilon^-\big(p_{k-2}(\mathbf{x})\big) - \omega_\varepsilon^-\big(p_{k-1}(\mathbf{x})\big).$$

Taking the expectation with respect to $\pi_{k-1}(\cdot \mid \mathbf{x})$ and noting that:

$$\sum_{\mathbf{y}} \pi_{k-1}(\mathbf{y} \mid \mathbf{x})\mathbf{1}_{\{r(\mathbf{x},\mathbf{y})=1\}} = p_{k-1}(\mathbf{x}) \tag{12}$$

$$\sum_{\mathbf{y}} \pi_{k-1}(\mathbf{y} \mid \mathbf{x})\mathbf{1}_{\{r(\mathbf{x},\mathbf{y})=0\}} = 1 - p_{k-1}(\mathbf{x}) \tag{13}$$

we obtain:

$$D_{\text{KL}}\big(\pi_{k-1}\|\pi_k\big) = \log \frac{Z_{k-1}(\mathbf{x})}{Z_{k-2}(\mathbf{x})} + \tfrac{1}{\beta}\big[\Delta_k^+(\mathbf{x})p_{k-1}(\mathbf{x}) - \Delta_k^-(\mathbf{x})(1 - p_{k-1}(\mathbf{x}))\big] \tag{14}$$

Substituting the definitions of $\Delta_k^+(\mathbf{x})$ and $\Delta_k^-(\mathbf{x})$ and expanding:

$$D_{\mathrm{KL}}\big(\pi_{k-1}\|\pi_k\big) = \log\frac{Z_{k-1}(\mathbf{x})}{Z_{k-2}(\mathbf{x})} + \tfrac{1}{\beta}\big[\omega_\varepsilon^+(p_{k-2}(\mathbf{x}))p_{k-1}(\mathbf{x}) - \omega_\varepsilon^+(p_{k-1}(\mathbf{x}))p_{k-1}(\mathbf{x}) \tag{15}$$

$$- \omega_\varepsilon^-(p_{k-2}(\mathbf{x}))(1 - p_{k-1}(\mathbf{x})) + \omega_\varepsilon^-(p_{k-1}(\mathbf{x}))(1 - p_{k-1}(\mathbf{x}))\big] \tag{16}$$

A key observation is that for any $p$, we have $\omega_\varepsilon^+(p)p - \omega_\varepsilon^-(p)(1 - p) = 0$, which can be verified from their definitions. Applying this identity to the terms involving $p_{k-1}(\mathbf{x})$:

$$\omega_\varepsilon^+(p_{k-1}(\mathbf{x}))p_{k-1}(\mathbf{x}) - \omega_\varepsilon^-(p_{k-1}(\mathbf{x}))(1 - p_{k-1}(\mathbf{x})) = 0$$

Therefore, these terms cancel out, yielding:

$$D_{\mathrm{KL}}\big(\pi_{k-1}(\cdot\mid\mathbf{x})\,\|\,\pi_k(\cdot\mid\mathbf{x})\big) = \frac{1}{\beta}\Big[\omega_\varepsilon^+\big(p_{k-2}(\mathbf{x})\big)p_{k-1}(\mathbf{x}) - \omega_\varepsilon^-\big(p_{k-2}(\mathbf{x})\big)\big(1 - p_{k-1}(\mathbf{x})\big)\Big] - \log\frac{Z_{k-2}(\mathbf{x})}{Z_{k-1}(\mathbf{x})}$$

which completes the proof. $\qquad\square$

## B.2 Integral Form of the Step-Wise Learnability Gap

According to the closed-form of the step-wise learnability derived in the previous section, we can further transform the difference of step-wise learnability into an integral form involving partial derivatives. Then we prove that these partial derivatives are positive, which establishes our main result.

We simplify the notation of the step-wise learnability in Lemma 3 as follows:

$$D(a, b) = \frac{1}{\beta}\Big[\omega_\varepsilon^+(b)a - \omega_\varepsilon^-(b)(1 - a)\Big] - \log\frac{Z(b)}{Z(a)}$$

where:

- $a$ represents the success probability at the current step
- $b$ represents the success probability at the previous step

Let $D_{k,\mathrm{GT}} = D(P_k^{\mathrm{GT}}, P_{k-1}^{\mathrm{GT}})$ represents the step-wise learnability when training with ground truth rewards, while $D_{k,\mathrm{FN}} = D(P_k^{\mathrm{FN}}, P_{k-1}^{\mathrm{FN}})$ represents the step-wise learnability when training with rewards containing false negatives.

**Lemma 4** (Integral Form of the Step-Wise Learnability Gap). *Let $\delta_k = D_{k,GT} - D_{k,FN}$ be the step-wise learnability gap at training step $k$, where $D_{k,GT}$ and $D_{k,FN}$ are defined in Equations equation 9 and equation 10. We can express $\delta_k$ as:*

$$\boxed{\delta_k = \int_0^{\Delta_k}[\partial_1 D + \partial_2 D](P_k^{GT} - t, P_{k-1}^{GT} - t)\, dt}$$

*where $\partial_1 D$ denotes $\frac{\partial D(a,b)}{\partial a}$, $\partial_2 D$ denotes $\frac{\partial D(a,b)}{\partial b}$, and $\Delta_k = P_k^{GT} - P_k^{FN} > 0$ by Lemma 1.*

*Proof.* We define a function $f(t) = D(P_k^{\mathrm{GT}} - t, P_{k-1}^{\mathrm{GT}} - t)$ for $t \in [0, \Delta_k]$. At the boundaries of the integration domain, we have:

$$f(0) = D(P_k^{\mathrm{GT}}, P_{k-1}^{\mathrm{GT}}) = D_{k,\mathrm{GT}} \tag{17}$$

At $t = \Delta_k = P_k^{\mathrm{GT}} - P_k^{\mathrm{FN}}$, we have:

$$f(\Delta_k) = D(P_k^{\mathrm{FN}}, P_{k-1}^{\mathrm{FN}}) = D_{k,\mathrm{FN}} \tag{18}$$

Therefore, the learnability gap can be expressed as $\delta_k = f(0) - f(\Delta_k)$. By the fundamental theorem of calculus:

$$\delta_k = -\int_0^{\Delta_k} f'(t)\, dt$$

Computing $f'(t)$ via the chain rule:

$$f'(t) = \frac{d}{dt} D(P_k^{\mathrm{GT}} - t, P_{k-1}^{\mathrm{GT}} - t) \tag{19}$$

$$= \partial_1 D(P_k^{\mathrm{GT}} - t, P_{k-1}^{\mathrm{GT}} - t) \cdot (-1) + \partial_2 D(P_k^{\mathrm{GT}} - t, P_{k-1}^{\mathrm{GT}} - t) \cdot (-1) \tag{20}$$

$$= -[\partial_1 D + \partial_2 D](P_k^{\mathrm{GT}} - t, P_{k-1}^{\mathrm{GT}} - t) \tag{21}$$

Therefore:

$$\delta_k = \int_0^{\Delta_k} [\partial_1 D + \partial_2 D](P_k^{\mathrm{GT}} - t, P_{k-1}^{\mathrm{GT}} - t)\, dt$$

$$\square$$

Since $\Delta_k > 0$ by Lemma 1, proving $\delta_k > 0$ reduces to showing that the integrand $[\partial_1 D + \partial_2 D](a, b) > 0$ throughout the integration domain. In other words, if the sum of partial derivatives of $D$ with respect to its arguments is positive, then the step-wise learnability with ground truth rewards exceeds that with false negative rewards.

**Lemma 5** (Positivity of the Partial Derivatives)**.** *For any $(a, b) \in (0, 1)^2$ satisfying $b < a < 2b$, the following inequality holds:*

$$\boxed{[\partial_1 D + \partial_2 D](a, b) > 0}$$

*Proof.* We begin by computing the partial derivatives of the function

$$D(a, b) = \frac{1}{\beta}\big(W^+(b)\, a - W^-(b)\,(1 - a)\big) - \log \frac{Z(b)}{Z(a)},$$

where we use $W^+$ and $W^-$ as shorthand for $\omega_\varepsilon^+$ and $\omega_\varepsilon^-$ to simplify notation.

Direct differentiation with respect to $a$ and $b$ yields:

$$\partial_1 D(a, b) = \frac{1}{\beta}\big(W^+(b) + W^-(b)\big) + \frac{Z'(a)}{Z(a)},$$

$$\partial_2 D(a, b) = \frac{1}{\beta}\big(a\, W^{+\prime}(b) - (1 - a)\, W^{-\prime}(b)\big) - \frac{Z'(b)}{Z(b)}.$$

Summing these two partial derivatives, we obtain:

$$[\partial_1 D + \partial_2 D](a, b) = \underbrace{\frac{1}{\beta} T(b)}_{A} + \underbrace{\left[\frac{Z'(a)}{Z(a)} - \frac{Z'(b)}{Z(b)}\right]}_{B},$$

where

$$T(b) = W^+(b) + W^-(b) + aW^{+\prime}(b) - (1 - a)W^{-\prime}(b).$$

Our proof strategy is to show that both term $A$ and term $B$ are positive under the given conditions.

**Part A: Proving $\frac{1}{\beta} T(b) > 0$**

Recall the definitions:

$$W^+(p) = \frac{1 - p}{\sqrt{p(1 - p)} + \varepsilon}, \quad W^-(p) = \frac{p}{\sqrt{p(1 - p)} + \varepsilon},$$

where $\varepsilon > 0$ is a small positive constant, and denote $d(b) = \sqrt{b(1-b)} + \varepsilon$.

For the term $T(b) = W^+(b) + W^-(b) + aW^{+'}(b) - (1-a)W^{-'}(b)$, after simplification, we have:

$$T(b) = \frac{d(b) - d'(b)(a - b)}{d(b)^2}$$

where $d'(b) = \frac{1-2b}{2\sqrt{b(1-b)}}$. Thus

$$T(b) = \frac{b(1-b) + \varepsilon\sqrt{b(1-b)} - (a-b)(1-2b)}{\sqrt{b(1-b)}(\sqrt{b(1-b)} + \varepsilon)^2}$$

For $b > \frac{1}{2}$, since $1 - 2b < 0$, $a - b > 0$, we have $T(b) > 0$.

For $b \le \frac{1}{2}$, by using $a < 2b$:

$$T(b) > \frac{b(1-b) - b(1-2b)}{\sqrt{b(1-b)}(\sqrt{b(1-b)} + \varepsilon)^2} > \frac{b^2}{\sqrt{b(1-b)}(\sqrt{b(1-b)} + \varepsilon)^2} > 0$$

Thus, $T(b) > 0$ for all $b \in (0,1)$, which implies $\frac{1}{\beta}T(b) > 0$.

**Part B: Proving $\frac{Z'(a)}{Z(a)} - \frac{Z'(b)}{Z(b)} > 0$ when $a > b$**

We want to prove that $g(p) = \frac{Z'(p)}{Z(p)}$ is strictly increasing, which will show that when $a > b$, we have $g(a) - g(b) > 0$.

Recall that:

$$Z(p) = P_{\text{ref}}e^{u(p)} + (1 - P_{\text{ref}})e^{-v(p)} \tag{22}$$

$$u(p) = \frac{1}{\beta}W^+(p) \tag{23}$$

$$v(p) = \frac{1}{\beta}W^-(p) \tag{24}$$

Define the weights:

$$w_1(p) = \frac{P_{\text{ref}}e^{u(p)}}{Z(p)}, \quad w_2(p) = \frac{(1 - P_{\text{ref}})e^{-v(p)}}{Z(p)} \tag{25}$$

Note that $w_1(p) + w_2(p) = 1$.

The derivative of $Z(p)$ is:

$$Z'(p) = P_{\text{ref}}e^{u(p)}u'(p) + (1 - P_{\text{ref}})e^{-v(p)}(-v'(p)) \tag{26}$$
$$= Z(p) \cdot [w_1(p)u'(p) - w_2(p)v'(p)] \tag{27}$$

Therefore:

$$g(p) = \frac{Z'(p)}{Z(p)} = w_1(p)u'(p) - w_2(p)v'(p) \tag{28}$$

Thus we have

$$g'(p) = w_1'(p)u'(p) + w_1(p)u''(p) + w_2'(p)(-v'(p)) + w_2(p)(-v''(p)) \tag{29}$$

We know that $w_1(p) + w_2(p) = 1$, so $w_1'(p) + w_2'(p) = 0$, i.e., $w_1'(p) = -w_2'(p)$.

Using the definition of $w_1(p)$ and $w_2(p)$, we can derive:

$$w_1'(p) = w_1(p)[u'(p) - g(p)] \tag{30}$$
$$w_2'(p) = w_2(p)[(-v'(p)) - g(p)] \tag{31}$$

Substituting these into the expression for $g'(p)$:

$$g'(p) = w_1(p)[u'(p) - g(p)]u'(p) + w_2(p)[(-v'(p)) - g(p)](-v'(p)) \tag{32}$$
$$= w_1(p)u'(p)^2 - w_1(p)g(p)u'(p) + w_2(p)v'(p)^2 - w_2(p)g(p)(-v'(p)) \tag{33}$$
$$= w_1(p)u'(p)^2 + w_2(p)v'(p)^2 - g(p)^2 \tag{34}$$

We can expand $g(p)^2$ as:

$$g(p)^2 = w_1(p)^2 u'(p)^2 - 2w_1(p)w_2(p)u'(p)v'(p) + w_2(p)^2 v'(p)^2 \tag{35}$$

Substituting this into our expression for $g'(p)$:

$$g'(p) = w_1(p)u'(p)^2 + w_2(p)v'(p)^2 - [w_1(p)^2 u'(p)^2 - 2w_1(p)w_2(p)u'(p)v'(p) + w_2(p)^2 v'(p)^2] \tag{36}$$
$$= w_1(p)u'(p)^2(1 - w_1(p)) + w_2(p)v'(p)^2(1 - w_2(p)) + 2w_1(p)w_2(p)u'(p)v'(p) \tag{37}$$
$$= w_1(p)w_2(p)u'(p)^2 + w_1(p)w_2(p)v'(p)^2 + 2w_1(p)w_2(p)u'(p)v'(p) \tag{38}$$
$$= w_1(p)w_2(p)[u'(p) + v'(p)]^2 \tag{39}$$

This is positive since it's a squared term multiplied by positive weights ($w_1(p) > 0$ and $w_2(p) > 0$).

Consequently, $g(p) = \frac{Z'(p)}{Z(p)}$ is strictly increasing, which means that when $a > b$, we have $g(a) - g(b) > 0$.

Combining the results from **Part A** and **Part B**, we have:

$$[\partial_1 D + \partial_2 D](a, b) = \frac{1}{\beta}T(b) + \left[\frac{Z'(a)}{Z(a)} - \frac{Z'(b)}{Z(b)}\right] > 0$$

This completes the proof of Lemma 5. $\square$

### B.3 Proof of Theorem 1

Having established the necessary lemmas, we now complete the proof of Theorem 1.

*Proof.* From Lemma 4, we have expressed the step-wise learnability gap as an integral:

$$\delta_k = \int_0^{\Delta_k} [\partial_1 D + \partial_2 D](P_k^{\text{GT}} - t, P_{k-1}^{\text{GT}} - t)\,dt$$

where $\Delta_k = P_k^{\text{GT}} - P_k^{\text{FN}} > 0$ by Lemma 1.

From Lemma 5, we have established that $[\partial_1 D + \partial_2 D](a, b) > 0$ for all pairs $(a, b) \in (0, 1)^2$ satisfying $b < a < 2b$ (Assumption 1 and 2).

Since the integrand $[\partial_1 D + \partial_2 D](P_k^{\text{GT}} - t, P_{k-1}^{\text{GT}} - t)$ is positive throughout the integration domain, and the integration is performed over a positive interval $[0, \Delta_k]$, we conclude that $\delta_k > 0$ for all $k$.

$\square$

This theoretical result highlights the importance of accurate reward signals in reinforcement learning. False negatives in reward feedback significantly impede the learning process by reducing the step-wise improvement of the policy at each iteration, potentially leading to slower convergence and suboptimal performance.

**Note on False Positives.** We note that analogously defining a false-positive reward model and applying the same framework does not yield the conclusion that the learnability of FP training exceeds that of ground-truth training. Under such a relabeling, the required monotonicity condition (Assumption 1) no longer holds for the relabeled quantities. Empirically, Appendix E.6 shows that while false positives can impair final performance, they do not significantly slow convergence relative to the cleaner ground-truth reward setting. This explains why false negatives are particularly damaging to RL training efficiency, whereas the practical impact of false positives remains comparatively limited.

## C  More on Experimental Setups

In this section, we detail our setups for the experiments.

### C.1  Experimental Setups for Zero RL

We follow Luo et al. (2025) and use the following hyper-parameters detailed in Table 3 for Zero RL training. We perform experiments on 8 A100 GPUs. The model is trained using VERL Sheng et al. (2025).

Table 3: This table shows the hyper-parameters for zero RL training.

| Hyper-parameter | Value |
|---|---|
| Learning Rate | $1 \times 10^{-6}$ |
| Number of Epochs | 12 |
| Number of Devices | 8 |
| Rollout Batch Size | 128 |
| PPO Mini Batch Size | 64 |
| Max Prompt Length | 1024 |
| Max Response Length | 3072 (Qwen2.5-Math-7B), 4096 (Qwen2.5-7B) |
| KL Coefficient | 0.001 |
| Rollout Engine | vllm (v0.8.2) |
| Optimizer | Adamw |
| Learning Rate Scheduler | cosine |
| Warmup Ratio | 0.1 |
| Max Sequence Length | 4096 |

### C.2  TinyV Data Curation

**Real Example Generation.** We utilize the *seemingly incorrect prompt-response pairs* collected in Section 4 as the source of real examples. Specifically, for each prompt-response pair $(\mathbf{x}, \mathbf{y}_i)$ marked as incorrect by *Prime Verifier*, we adopt LLM annotations as the ground truth label: "True" for a response that is correct and "False" otherwise. Additionally, we retain the intermediate analysis of LLMs for TinyV-Think training.

**Synthetic Example Generation.** To enhance coverage, ensure robustness, and balance the dataset with an equal number of "True" and "False" labels, we augment the dataset with synthetically generated false negatives. Specifically, we prompt Qwen2.5-72B-Instruct to generate potential false negative cases for a given question by introducing variations such as LaTeX formatting differences, numerical approximations, or alternative mathematical expressions that preserve semantic equivalence. These generated candidates are then re-annotated by LLMs to confirm their correctness. As with the real examples, we retain the intermediate analysis of LLMs. The prompts used for generating synthetic examples are provided in Appendix G.3.

Table 4: This table shows the hyper-parameters for supervised fine-tuning of TINYV.

| Hyper-parameter | Value |
|---|---|
| Learning Rate | $1 \times 10^{-5}$ |
| Number of Epochs | 2 |
| Number of Devices | 8 |
| Per-device Batch Size | 8 |
| Gradient Accumulation Steps | 8 |
| Effective Batch Size | 512 |
| Optimizer | `Adamw` |
| Learning Rate Scheduler | `cosine` |
| Warmup Ratio | 0.1 |
| Max Sequence Length | 4096 |

**Prompt Template for TinyV Training and Inference**

```
You are an AI tasked with identifying false negatives in answer verification. A false
    negative occurs when a model's answer is essentially correct but is marked as
    incorrect due to minor discrepancies or formatting issues. Your job is to analyze the
    given question, ground truth answer, and model answer to determine if the model's
    answer is actually correct despite appearing different from the ground truth.

<question>{{QUESTION}}</question>

<ground_truth_answer>{{GROUND_TRUTH_ANSWER}}</ground_truth_answer>

<model_answer>{{MODEL_ANSWER}}</model_answer>

Return "True" if the model's answer is correct, otherwise return "False".
```

Figure 8: Prompt Template for TINYV Training and Inference.

### C.3 TinyV Training

Table 4 demonstrates the detailed supervised fine-tuning (SFT) hyper-parameters for training TINYV. We perform experiments on 8 A100 GPUs. The training and inference template is demonstrated in Figure 8. The model is trained using Llama Factory Zheng et al. (2024).

### C.4 TinyV in Verifier-Centric Evaluation

To provide a stronger verifier-centric evaluation, we benchmark TINYV on all labeled examples from the *HardVerify-Math* benchmark. Following standard practice, we report TP, FN, FP, TN, Precision, Recall, F1, and Accuracy. We compare TINYV against three categories of baselines: (i) closed-source LLM judges (Grok-3-Mini, GPT-4o-mini), (ii) open-weight LLM judges (Qwen2.5-72B-Instruct, MiMo-V2-Flash), and (iii) purpose-built learned verifiers from the xVerify family (1.5B/7B/14B) Chen et al. (2025a).

As shown in Table 5, TINYV achieves strong overall performance. Notably:

- **TinyV outperforms all learned verifier baselines**, including the xVerify family across all sizes. TINYVachieves 89.4% accuracy, surpassing xVerify-1.5B (81.0%), xVerify-7B (86.8%), and xVerify-14B (86.0%).

- **TinyV achieves the highest recall among non-Grok baselines.** Since the core goal of TINYV is to recover correct responses wrongly rejected by rule-based verifiers, high recall is the most mission-critical metric. In contrast, xVerify models exhibit relatively low recall (0.71–0.78).

- **TinyV approaches the quality of much larger LLM judges at a fraction of the cost.** Only Grok-3-Mini surpasses TINYV in accuracy (0.9820 vs. 0.8940), while TINYV matches or exceeds Qwen2.5-72B-Instruct (0.9120) and GPT-4o-mini (0.8920) despite being 40–50× smaller.

| Verifier | TP | FN | FP | TN | Precision | Recall | F1 | Acc |
|---|---|---|---|---|---|---|---|---|
| Grok-3-Mini | 246 | 4 | 5 | 244 | **0.9801** | **0.9840** | **0.9820** | **0.9820** |
| Qwen2.5-72B-Instruct | 243 | 7 | 37 | 213 | 0.8679 | 0.9720 | 0.9170 | 0.9120 |
| **TinyV(ours)** | 246 | 4 | 49 | 201 | 0.8339 | **0.9840** | 0.9027 | 0.8940 |
| GPT-4o-mini | 231 | 19 | 35 | 215 | 0.8684 | 0.9240 | 0.8953 | 0.8920 |
| xVerify-7B-I | 192 | 58 | 8 | 242 | 0.9600 | 0.7680 | 0.8533 | 0.8680 |
| xVerify-14B-Ia | 194 | 56 | 14 | 236 | 0.9327 | 0.7760 | 0.8472 | 0.8600 |
| MiMo-V2-Flash | 248 | 2 | 94 | 156 | 0.7251 | 0.9920 | 0.8378 | 0.8080 |
| xVerify-1.5B-I | 177 | 73 | 22 | 228 | 0.8894 | 0.7080 | 0.7884 | 0.8100 |

Table 5: Verifier-centric evaluation on labeled examples from *HardVerify-Math*.

## D  HardVerify-Math Benchmark

In this section, we detail our *HardVerify-Math Bench*, a benchmark comprising 250 hard-to-verify answers that span all categories and the taxonomy discussed in Section 4. The dataset consists of two parts: (1) from existing benchmarks, we manually select 115 questions from the Olympiad benchmark and 10 questions from the MATH test sets, which are prone to false negatives due to their complex answer formats; (2) from other sources, we include 125 questions from the *Big-Math* dataset, selected based on a LLaMA-3.1-8B pass rate of less than 0.05 and identified as challenging to verify by human experts. Each question in *HardVerify-Math Bench* results in at least one false negative when evaluated using *Prime Verifier*. We include the incorrect answer that triggers the false negative, along with the question and its ground truth answer, for reference. Figure 9 illustrates examples, while Figure 10 shows the sources of the questions.

*HardVerify-Math Bench* is compact by design but offers broad coverage and direct FN evaluation. Each example requires dual-LLM annotation and expert verification, making large-scale construction prohibitively expensive. We therefore prioritize **diverse coverage** over raw scale, which allows *HardVerify-Math Bench* to remain cost-controlled while still providing broad coverage of verifier failure modes during model evaluation.

## E  More Experimental Results

### E.1  Comparison Against Rule-based Verifiers

Figure 11 compares the training dynamics and final performance of TINYV, TINYV-THINK, *Math-Verify*, and *Prime Verifier* on AMC, MATH, Olympiad, and the *HardVerify-Math* benchmark. All models are trained on QWEN2.5-MATH-7B under GRPO with identical GRPO settings.

Across all benchmarks, TINYV and TINYV-THINK consistently achieve higher scores and faster convergence than rule-based verifiers. In particular, the smoothed learning curves show that TINYV-based verifiers reach strong performance earlier in training, while *Math-Verify* and *Prime Verifier* exhibit slower and noisier convergence, reflecting their high false-negative rates during reward calculation.

Although TINYV-THINK exhibits slightly stronger asymptotic performance on some benchmarks (e.g., AMC and Olympiad), its computational cost is substantially higher. Training on TINYV-THINK requires 53.73 hours, compared to only 18.71 hours for TINYV, making it significantly less efficient. As a result, we adopt TINYV as the default verifier in all main experiments.

**Example 1 (Olympiad Benchmark)**

```
Question: Determine all real numbers $x>0$ for which\n\n$$\n\\log _{4} x-\\log _{x} 16=\\
    frac{7}{6}-\\log _{x} 8\n$$
Ground Truth: $2^{-2 / 3}$, $8$
Model Output: 8, \\frac{1}{\\sqrt[3]{4}}
```

**Example 2 (Olympiads Big-Math)**

```
Question: Which clock shows the correct time more often: one that is one minute slow or one
    that is stopped?
Ground Truth: A stopped clock shows the correct time more often.
Model Output: \\text{stopped}
```

**Example 3 (CN_K12)**

```
Question: After the epidemic, the tourism market in Yunnan has shown strong recovery this
    year. A travel agency rented two types of rooms, $A$ and $B$, during the Spring
    Festival this year. The number of rooms rented for $A$ at $4800$ yuan is the same as
    the number of rooms rented for $B$ at $4200$ yuan. The rent for each $A$ room this
    year is $30$ yuan more than the rent for each $B$ room. Find the rent for each $A$ and
     $B$ room this year.
Ground Truth: The rent for each $A$ room is $240$ yuan, and for each $B$ room is $210$ yuan
    .
Model Output: 240 \\text{ yuan (A)},\\ 210 \\text{ yuan (B)}
```

**Example 4 (ORCA Math)**

```
Question: A can do a piece of work in 12 days and B alone can do it in 14 days. How much
    time will both take to finish the work together?
Ground Truth: 6.46
Model Output: \\dfrac{84}{13}\\text{ days}
```

Figure 9: *HardVerify-Math Bench* Examples.

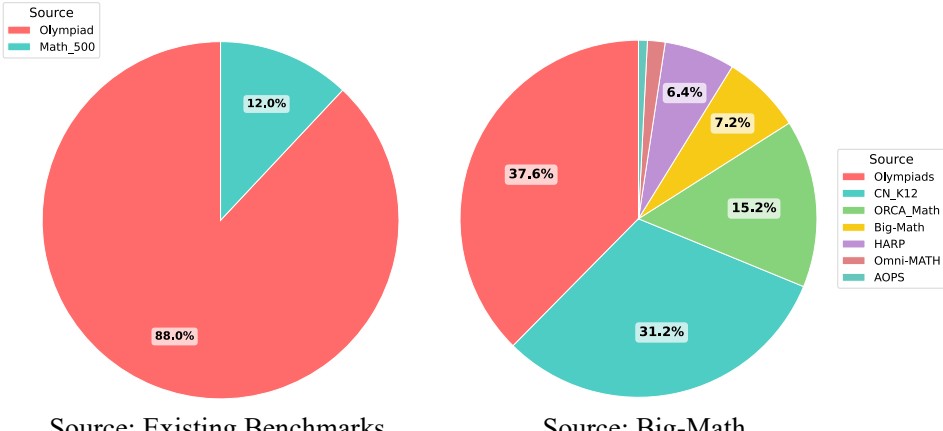

Source: Existing Benchmarks      Source: Big-Math

Figure 10: This figure shows the source distribution of *HardVerify-Math Bench*.

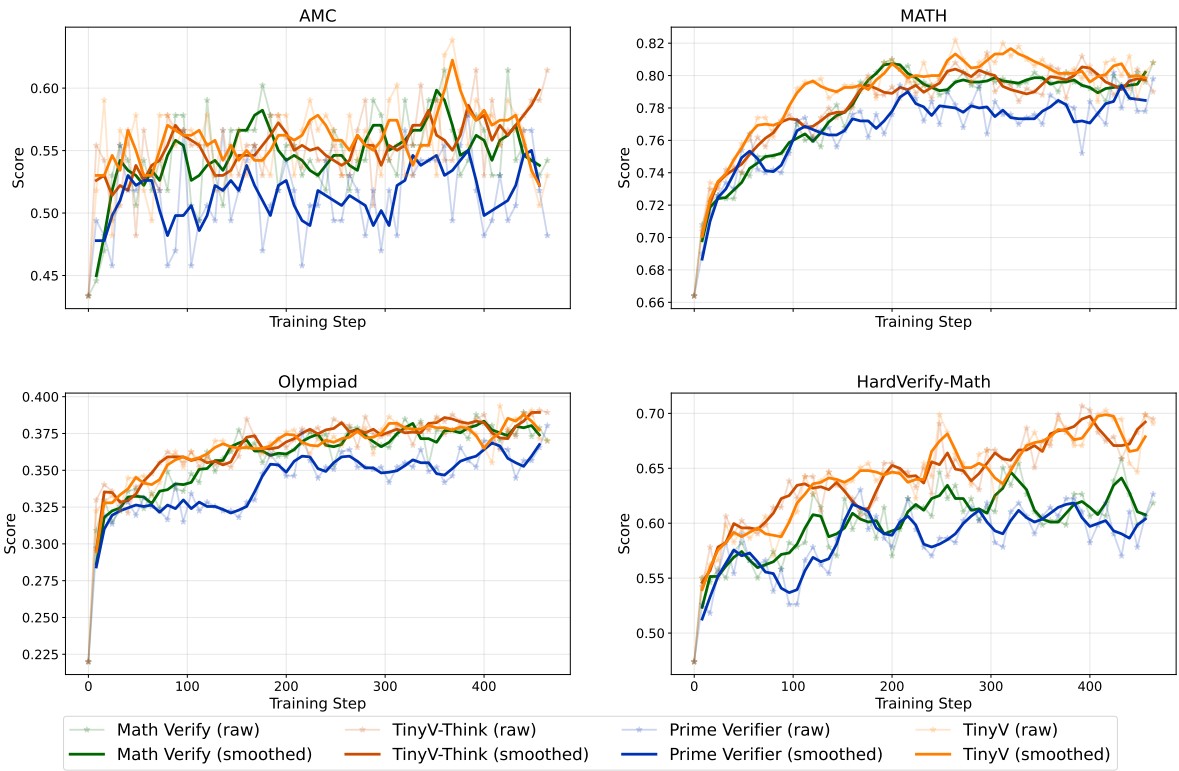

Figure 11: Training curves comparing TinyV, TinyV-Think, *Math-Verify*, and *Prime Verifier* on AMC, MATH, Olympiad, and HardVerify-Math. All methods use Qwen2.5-Math-7B as the base model.

Table 6 reports the final benchmark performance after training with different verifiers, including Math, Math-Ruler (hiyouga, 2025), *Prime Verifier*, and Math-Verify. Consistent with the learning curves in Figure 11, both TinyV and TinyV-Think outperform all rule-based verifiers across HardVerify, MATH, AMC, and Olympiad, with TinyV-Think achieving the highest overall average.

| Verifier | HardVerify | MATH | AMC | Olympiad | Average |
|----------|-----------|------|-----|----------|---------|
| Qwen2.5-Math-7B | 50.6% | 67.8% | 48.1% | 23.6% | 47.5% |
| Prime Verifier | 62.7% | 79.8% | 48.2% | 38.0% | 57.2% |
| Math | 61.0% | 77.8% | 50.6% | 36.1% | 56.4% |
| MathRuler | 59.0% | 77.4% | 49.3% | 36.4% | 55.5% |
| Math-Verify | 61.8% | 78.6% | 54.2% | 37.0% | 57.9% |
| **TinyV** | **69.1%** | **80.8%** | 53.0% | 37.0% | 60.0% |
| **TinyV-Think** | 68.1% | 79.0% | **61.4%** | **38.9%** | **61.9%** |

Table 6: Final benchmark performance (%) of different verifiers on Qwen2.5-Math-7B. TinyV and TinyV-Think consistently outperform all rule-based verifiers across all benchmarks.

### E.2 Comparison Against LLM-Judge Baselines

To further contextualize TinyV's practical value beyond rule-based verifiers, we conduct a direct comparison against strong LLM-judge-based RLVR baselines. Specifically, we evaluate three representative LLM judges, GPT-4o-mini (closed-source), DeepSeek-V3.2 (open-weight), and MiMo-V2-Flash (open-weight), as drop-in replacements for TinyV in the same add-on verification pipeline. All judges are constrained to produce structured True/False outputs using an identical prompt template to ensure a fair comparison. As shown in Table 7, TinyV achieves competitive performance while being significantly more efficient.

| Verifier (LLM-as-Judge, structured output) | MATH | AMC | Olympiad |
|---|---|---|---|
| Qwen2.5-7B | 46.8% | 25.3% | 22.1% |
| GPT-4o-mini | 75.0% | 43.4% | 32.4% |
| DeepSeek-V3.2 | 73.2% | 44.6% | 33.6% |
| **TinyV-1.5B (ours)** | **73.4%** | **43.4%** | **32.4%** |
| MiMo-V2-Flash | 51.8% | 25.3% | 22.9% |

Table 7: Performance comparison against LLM-judge baselines on MATH, AMC, and Olympiad benchmarks.

We summarize our key findings below:

- **TinyV matches strong commercial judges at a fraction of the size and cost.** TINYV-1.5B achieves RL training quality within 1–2 percentage points of GPT-4o-mini and DeepSeek-V3.2, despite being one to two orders of magnitude smaller. At the same time, TINYV's reward-calculation time is 3–6× faster than the commercial API judges (8.6s vs. 28.7s / 52.6s per step).

- **Not every LLM judge is suitable as a drop-in verifier.** MiMo-V2-Flash fails to serve as a reliable verifier, yielding only marginal gains over the base model.

- **TinyV occupies a distinctive point in the cost-quality frontier.** It is the only verifier in this comparison that simultaneously (i) matches commercial-judge quality, (ii) runs locally with no API dependency, and (iii) has the lowest reward-calculation overhead.

### E.3 Training Cost Analysis

We expand the overhead analysis along two axes: (i) model scale, and (ii) escalation fraction from the rule-based verifier.

**(i) Overhead by model scale.** We measured the end-to-end wall-clock overhead of the TINYV add-on pipeline versus *Prime Verifier* alone across multiple base models. As shown in Table 8, the overhead remains consistently low (below 7%) across all scales, demonstrating the scalability and efficiency of TINYV.

| Base Model | Overhead vs. Prime Verifier |
|---|---|
| Qwen3-1.7B | +4.3% |
| Qwen2.5-3B | +6.6% |
| Qwen2.5-7B | +6.1% |
| DeepSeek-Math-7B-Instruct | +2.7% |

Table 8: End-to-end wall-clock overhead of TINYV across different base models.

**(ii) Overhead by escalation fraction.** In the add-on pipeline, TINYV is only invoked when *Prime Verifier* flags a rollout as incorrect. We tracked this escalation fraction throughout training on Qwen2.5-7B with GRPO and observed that it remains high across all stages: approximately 90% of rollouts escalate to TINYV early in training, gradually decreasing to roughly 70% by the end of training.

Despite this consistently high escalation rate, the wall-clock overhead stays within the low ranges reported above. This highlights the effectiveness of our design choice: a compact 1.5B verifier is efficient enough per call that even near-universal escalation results in negligible end-to-end cost.

Figure 12 further illustrates the per-step time cost during GRPO training. Overall, TINYV incurs only an average 6% additional computational overhead while delivering substantial improvements in training effectiveness and final performance.

### E.4 Generalizability of TinyV on Unseen Math Training Data

To evaluate the robustness of TINYV under distribution shift within the mathematical domain, we conduct an out-of-distribution (OOD) training experiment using the *DeepMath* dataset He et al. (2025). Specifically,

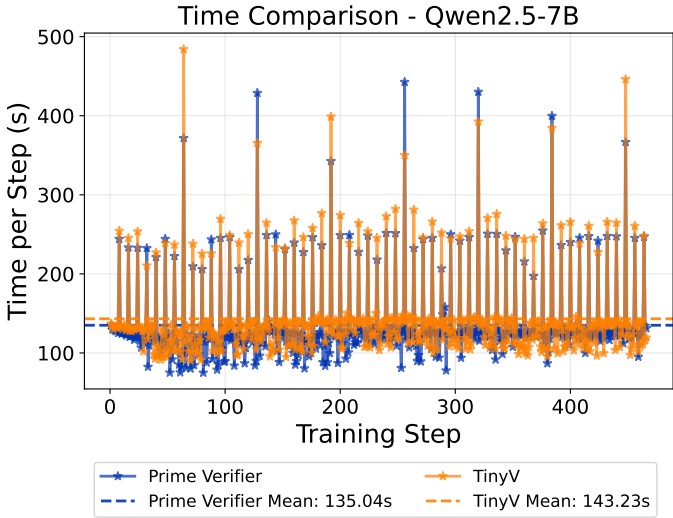

Figure 12: This figure compares the average time cost of TINYV with *Prime Verifier* during GRPO training. The peak occurs when saving model checkpoints.

we sample 5K hard questions (difficulty $\geq 7$) from DeepMath and perform the same GRPO training used in our main experiments.

We compare TINYV against the rule-based *Prime Verifier* under identical training settings. Table 9 reports the resulting performance. Despite being trained on a different data distribution, TINYV maintains its advantage over the rule-based verifier, improving the average benchmark score by approximately 1.1 percentage points. This result demonstrates that TINYV does not overfit to the training distribution of Big-Math-RL-Verified and generalizes robustly to unseen mathematical data.

| Experiment Setup | HardVerify | MATH | AMC | Olympiad | Average |
|---|---|---|---|---|---|
| Qwen2.5-Math-7B | 50.6% | 67.8% | 48.1% | 23.6% | 47.5% |
| Prime Verifier (DeepMath OOD RL) | 62.2% | 77.0% | 51.8% | 35.6% | 56.7% |
| **TinyV (DeepMath OOD RL)** | **62.7%** | **78.4%** | **54.2%** | **35.6%** | **57.7%** |

Table 9: Generalization of TINYV to unseen mathematical training data. Models are trained using 5K DeepMath questions (difficulty $\geq 7$) and evaluated on standard mathematical benchmarks.

### E.5 Generalizability of TinyV on Other Domains

While TINYV is primarily trained on mathematical reasoning data, it is designed to mitigate false negatives of rule-based verifiers and should therefore generalize to other domains where rule-based verifiers struggle. To evaluate this hypothesis, we construct a mixed-domain reinforcement learning setup by combining 5K samples from **Big-Math-RL-Verified** with 5K samples from the **Natural Reasoning** dataset Yuan et al. (2025a). The Natural Reasoning dataset consists of long-form, sentence-level answers across diverse reasoning tasks, which are notoriously difficult for rule-based verification systems.

We follow the same training configuration as in our main experiments and evaluate the resulting models on both our original mathematical benchmarks and **GPQA-Diamond** Rein et al. (2024), a highly challenging expert-curated benchmark containing 198 multiple-choice questions in biology, physics, and chemistry.

As shown in Table 10, integrating TINYVyields consistent improvements across all evaluation benchmarks. In particular, TINYVachieves a **5.6% absolute gain** on GPQA-Diamond over the Prime Verifier baseline (29.3% vs. 23.7%), despite being trained primarily on mathematical data. This result demonstrates that TINYV is transferred beyond mathematics and benefits in general.

| Experiment Setup | | HardVerify | MATH | AMC | Olympiad | GPQA-Diamond |
|---|---|---|---|---|---|---|
| Qwen2.5-Math-7B | – | 50.6% | 67.8% | 48.1% | 23.6% | 22.2% |
| Big-Math-RL-Verified | + Natural Reasoning (Prime) | 60.6% | 78.0% | 54.2% | 34.6% | 23.7% |
| | + Natural Reasoning (TinyV) | **64.7%** | **80.2%** | **56.6%** | **38.1%** | **29.3%** |

Table 10: Generalization of TinyVto non-mathematical domains. Models are trained on a mixture of Big-Math-RL-Verified and Natural Reasoning data and evaluated on both mathematical benchmarks and GPQA-Diamond.

### E.6 Impact of False Positives in RL Training

While reducing false negatives (FNs) is the primary goal of TinyV, false positives (FPs) where incorrect answers marked as correct may also affect reinforcement learning. To quantify this effect, we conduct a **controlled FP injection experiment** on Qwen2.5-Math-7B. Specifically, we randomly flip 20% of TinyV's zero-reward rollouts to positive reward, aligning the approximately 20% FN rate observed in Figure 4, and retrain the base model under the same RL tuning setups.

Table 11 reports the final benchmark performance under this perturbation. Despite the substantial level of injected false positives, the degradation in final performance is minimal. On HardVerify-Math, MATH and AMC, accuracy drops by only about 1-2%, while performance on Olympiad remains unchanged.

| Experiment Setup | HardVerify | MATH | AMC | Olympiad |
|---|---|---|---|---|
| Qwen2.5-Math-7B | 50.6% | 67.8% | 48.1% | 23.6% |
| TinyV + 20% FP | 67.9% | 78.4% | 51.8% | 37.0% |
| TinyV (no FP) | 69.1% | 80.8% | 53.0% | 37.0% |

Table 11: Effect of false positives (FPs) on RL training. We randomly flip 20% of zero-reward rollouts to positive reward.

We further analyze the training dynamics and find that the learning curves of TinyV and TinyV with 20% FPs overlap almost perfectly, indicating that false positives at this level do not slow convergence. In contrast, false negatives block reward entirely for correct answers, preventing learning on those questions. False positives only dilute the reward signal on questions that the model already solves, weakening but not eliminating the gradient. Consequently, reducing false negatives is substantially more critical for both training efficiency and final performance than minimizing false positives.

## F  Limitations and Broader Impacts

**Limitations.** This study primarily focuses on false negatives (FNs) in **rule-based final-answer verification for mathematical reasoning**. While we demonstrate the effectiveness of TinyV in this setting, FN patterns in other domains, such as theorem proving (Xin et al., 2024), code generation and software engineering (Wei et al., 2025), medical reasoning (Lai et al., 2025), or open-ended generation, may differ qualitatively. Consequently, the generalizability and effectiveness of TinyV in these domains remains to be studied.

Additionally, the current implementation of TinyV relies on *Prime Verifier*'s answer extraction mechanism (i.e., content within \boxed{}), which focuses solely on the final answer rather than the full reasoning trace. Extending TinyV to process and verify complete solution steps is an important direction for future work.

**Broader Impacts.** Our work advances the efficiency of reinforcement learning training for mathematical reasoning, potentially enhancing the efficiency of machine learning, without identified negative societal impacts.

---

**Prompt Template for False Negative Annotation (Part 1)**

```
## Task Description

You are an AI tasked with identifying false negatives in answer verification. A false negative occurs when a model'
    s answer is essentially correct but is marked as incorrect due to minor discrepancies or formatting issues.
    Your job is to analyze the given question, ground truth answer, and model answer to determine if the model's
    answer is actually correct despite appearing different from the ground truth.

Analyze the inputs carefully, considering the following:
1. Is the model's answer mathematically equivalent to the ground truth?
2. Are there minor formatting differences that don't affect the answer's correctness?
3. Is the model's answer more precise or in a different but valid format?

## Examples

Here are some examples of questions, ground truth answers, and model answers. All of them are correct.

**Example 1 (Order-Insensitive):**
<question>Determine all real values of $x$ for which $(x+8)^{4}=(2 x+16)^{2}$.</question>
<ground_truth_answer>-6,-8,-10</ground_truth_answer>
<model_answer>-10, -8, -6</model_answer>

<analysis>
```json
{
  "reasoning": "The model's answer lists the same values as the ground truth but in a different order. Since the
      question asks for all solutions, the order doesn't matter for correctness.",
  "is_correct": true
}
```
</analysis>

**Example 2 (Latex Expression):**
<question>A bag contains 3 green balls, 4 red balls, and no other balls. Victor removes balls randomly from the bag
    , one at a time, and places them on a table. Each ball in the bag is equally likely to be chosen each time
    that he removes a ball. He stops removing balls when there are two balls of the same colour on the table. What
     is the probability that, when he stops, there is at least 1 red ball and at least 1 green ball on the table
    ?</question>
<ground_truth_answer>$\\frac{4}{7}$</ground_truth_answer>
<model_answer>4/7</model_answer>

<analysis>
```json
{
  "reasoning": "The model's answer '4/7' is mathematically equivalent to the ground truth answer '$\\frac{4}{7}$'.
      The only difference is in the notation - the ground truth uses LaTeX fraction notation while the model uses
       a simple division format. The numerical value is identical in both cases."
  "is_correct": true
}
```
</analysis>
```

Figure 13: Prompt Template for Labeling FN Responses (Part 1)

## G   Prompt Templates

### G.1   Prompt for FN Annotation

Figure 13-14 demonstrates the prompt template for labeling false negative responses.

### G.2   Prompt for FN Category Annotations

Figure 15-17 demonstrates the prompt template for labeling FN categories.

### G.3   Prompt for Generating Synthetic FN Examples

Figure 18 demonstrates the prompt template for generating Synthetic FN Examples.

---

**Prompt Template for False Negative Annotation (Part 2)**

```
**Example 3 (Variable):**
<question>If $T=x^{2}+\\frac{1}{x^{2}}$, determine the values of $b$ and $c$ so that $x^{6}+\\frac{1}{x^{6}}=T^{3}+
    b T+c$ for all non-zero real numbers $x$.</question>
<ground_truth_answer>-3,0</ground_truth_answer>
<model_answer>b=-3, c=0</model_answer>

<analysis>
```json
{
  "reasoning": "The model's answer 'b=-3, c=0' is mathematically equivalent to the ground truth answer '-3,0'. The
      model simply labeled the values with their corresponding variables, which provides more clarity but doesn't
      change the mathematical content of the answer.",
  "is_correct": true
}
```
</analysis>

**Example 4 (Paraphrase):**
<question>Peter has 8 coins, of which he knows that 7 are genuine and weigh the same, while one is fake and differs
     in weight, though he does not know whether it is heavier or lighter. Peter has access to a balance scale,
    which shows which side is heavier but not by how much. For each weighing, Peter must pay Vasya one of his
    coins before the weighing. If Peter pays with a genuine coin, Vasya will provide an accurate result; if a fake
     coin is used, Vasya will provide a random result. Peter wants to determine 5 genuine coins and ensure that
    none of these genuine coins are given to Vasya. Can Peter guaranteedly achieve this?</question>
<ground_truth_answer>Petya can guarantee finding 5 genuine coins.</ground_truth_answer>
<model_answer>Yes, Peter can guarantee finding 5 genuine coins while ensuring that none of these genuine coins are
    paid to Vasya.</model_answer>

<analysis>
```json
{
  "reasoning": "The model's answer correctly states that Peter can guarantee finding 5 genuine coins, which matches
      the ground truth. The model provides additional details about ensuring none of these coins are paid to Vasya
      , but this doesn't change the correctness of the answer."
  "is_correct": true
}
```
</analysis>

## Input

Now, please analyze the following question, ground truth answer, and model answer.

<question>
{{QUESTION}}
</question>

<ground_truth_answer>
{{GROUND_TRUTH_ANSWER}}
</ground_truth_answer>

<model_answer>
{{MODEL_ANSWER}}
</model_answer>

## Output

Please provide your analysis in the following JSON format:
<analysis>
```json
{
  "reasoning": "Your detailed reasoning here",
  "is_correct": true/false
}
```
</analysis>

Ensure your reasoning is thorough and considers all aspects of the answers. The "is_correct" field should be true
    if the model's answer is essentially correct despite any minor differences from the ground truth and false
    otherwise.
```

Figure 14: Prompt Template for Labeling FN Responses (Part 2)

---

**Prompt Template for Labeling FN Categories (Part 1)**

```
## Task Description

You are an AI assistant tasked with classifying schemes for common types of equivalence and mismatch between
    mathematical answers.

## Taxonomy

---

### 1. Formatting and Syntax Differences

Differences in formatting and/or syntax that do not affect mathematical meaning.

* **1.1 Formatting -> Whitespace and Spacing Issues**
    * *Description:* Variations in spaces around operators, within expressions, or between elements.
    * *Example:* `ground truth answer`: `f(x) = 2 x`, `model answer`: `f(x)=2x`
* **1.2 Formatting -> Symbol Representation Issues**
    * *Description:* Differences in symbol notation, including Unicode vs. command-based symbols, delimiter styles,
        or minor symbol variations (e.g., degree symbols, infinity notation).
    * *Example:* `ground truth answer`: `$(-\infty,-3)\cup(3,+\infty)$`, `model answer`: `$(-\infty,-3)\cup(3,\infty
        )$`
* **1.3 Formatting -> Markup Variation Issues**
    * *Description:* Differences in syntax for equivalent rendering, such as LaTeX command choices or delimiter
        sizing.
    * *Example:* `ground truth answer`: `\frac{32}{9}`, `model answer`: `\dfrac{32}{9}`
* **1.4 Formatting -> Unit Representation Issues**
    * *Description:* Differences in the inclusion, omission, or representation of units (e.g., missing units,
        abbreviated vs. full unit names).
    * *Example:* `ground truth answer`: `18.8^\circ`, `model answer`: `18.8`
* **1.5 Formatting -> Contextual Addition or Omission Issues**
    * *Description:* Missing or extra prefixes (e.g., "x=") or explanatory text not affecting the core answer,
        excluding units.
    * *Example:* `ground truth answer`: `N=n`, `model answer`: `n`
* **1.6 Formatting -> Other Formatting Issues**
    * *Description:* Miscellaneous formatting differences, such as newline characters or non-alphanumeric separators
        .
    * *Example:* `ground truth answer`: `60^\textcirc 42'`, `model answer`: `60^\circ 42'`

---

### 2. Mathematical Notation Variations

Differences in standard mathematical conventions for expressing the same concept.

* **2.1 Notation -> Interval vs. Inequality Notation**
    * *Description:* Representing ranges as intervals or inequalities.
    * *Example:* `ground truth answer`: `(-\infty, -5)`, `model answer`: `k < -5`
* **2.2 Notation -> Ratio and Proportion Variations**
    * *Description:* Different ways of expressing ratios or proportions (e.g., colon, fraction, or single value).
    * *Example:* `ground truth answer`: `2:1`, `model answer`: `2/1`
* **2.3 Notation -> Aggregated vs. Individual Solution Variations**
    * *Description:* Using symbols like $\pm$ or listing solutions separately.
    * *Example:* `ground truth answer`: `1 $\pm$ \sqrt{19}`, `model answer`: `1 + \sqrt{19}, 1 - \sqrt{19}`
* **2.4 Notation -> Vector and Matrix Notation Variations**
    * *Description:* Variations in displaying vectors or matrices.
    * *Example:* `ground truth answer`: `\begin{pmatrix} -7 \\ 16 \\ 5 \end{pmatrix}`, `model answer`: `(-7,16,5)`
* **2.5 Notation -> Other Notation Variations**
    * *Description:* Variations due to regional conventions (e.g., decimal points vs. commas) or other notation
        differences.
    * *Example:* `ground truth answer`: `3.14`, `model answer`: `3,14`

---
```

Figure 15: Prompt Template for Labeling FN Categories (Part 1)

**Prompt Template for Labeling FN Categories (Part 2)**

```
### 3. Mathematical Expression Equivalencies

Expressions that differ in form but are mathematically equivalent.

* **3.1 Expression -> Algebraic Equivalence Variations**
    * *Description:* Different but equivalent algebraic forms, including term ordering, factoring, or simplification
       .
    * *Example:* `ground truth answer`: `\frac{1-p^{2}}{3}`, `model answer`: `\frac{-p^2+1}{3}`
* **3.2 Expression -> Root and Exponent Form Variations**
    * *Description:* Using roots, fractional exponents, or simplified exponents differently.
    * *Example:* `ground truth answer`: `2^{-2 / 3}`, `model answer`: `\frac{1}{\sqrt[3]{4}}`
* **3.3 Expression -> Logarithmic and Trigonometric Form Variations**
    * *Description:* Equivalent forms using logarithmic or trigonometric identities.
    * *Example:* `ground truth answer`: `\frac{\log 2}{\log 2-\log 3}`, `model answer`: `-\frac{\ln 2}{\ln 3-\ln 2}`
* **3.4 Expression -> Other Equivalence Variations**
    * *Description:* Equivalencies in combinatorial quantities, complex numbers, or other mathematical structures.
    * *Example:* `ground truth answer`: `\frac{3 m}{2}-1`, `model answer`: `\dfrac{3m - 2}{2}`

---

### 4. Numerical Representation Differences

Variations in how numerical values are presented.

* **4.1 Numeric -> Exact vs. Approximate Form Variations**
    * *Description:* Exact (fraction, symbolic) vs. decimal or percentage approximations.
    * *Example:* `ground truth answer`: `\frac{600}{7}`, `model answer`: `85.71`
* **4.2 Numeric -> Alternative Exact Form Variations**
    * *Description:* Different exact representations, such as scientific notation or evaluated powers.
    * *Example:* `ground truth answer`: `10^{3}`, `model answer`: `1000`
* **4.3 Numeric -> Rounding and Precision Variations**
    * *Description:* Approximations with different decimal places or rounding rules.
    * *Example:* `ground truth answer`: `1.27\%`, `model answer`: `1.3\%`
* **4.4 Numeric -> Other Numerical Variations**
    * *Description:* Other numerical format differences, such as mixed vs. improper fractions.
    * *Example:* `ground truth answer`: `6\frac{1}{64}`, `model answer`: `6.015625`

---

### 5. Language and Contextual Variations

Differences in natural language or implied context.

* **5.1 Language -> Presence/Absence of Explanatory Text**
    * *Description:* Model output or ground truth includes additional descriptive text, or vice versa.
    * *Example:* `ground truth answer`: `10,11,12,13,14,-2,-1,0,1,2`, `model answer`: `Sequence 1: -2, -1, 0, 1, 2
        and Sequence 2: 10, 11, 12, 13, 14`
* **5.2 Language -> Implicit vs. Explicit Variable/Function Assignment**
    * *Description:* One output explicitly assigns values to variables or defines a function while the other lists
        values or the expression directly.
    * *Example:* `ground truth answer`: `16,3,1,1`, `model answer`: `w=16, d=3, a=1, b=1`
* **5.3 Language -> Phrasing and Conciseness Variations**
    * *Description:* Differences in wording, synonyms, or level of detail.
    * *Example:* `ground truth answer`: `\text{Any odd number of participants}`, `model answer`: `odd`
* **5.4 Language -> Other Language Variations**
    * *Description:* Minor differences in separators (e.g., "and" vs. comma) or answer structure.
    * *Example:* `ground truth answer`: `1,3`, `model answer`: `1 \text{ and } 3`

---
```

Figure 16: Prompt Template for Labeling FN Categories (Part 2)

---

**Prompt Template for Labeling FN Categories (Part 3)**

```
### 6. Set and List Differences

Variations in presenting collections of results, assuming correctness.

* **6.1 Set/List -> Order of Element Variations**
    * *Description:* Different sequencing of elements in sets or lists where order is not mathematically significant
        .
    * *Example:* 'ground truth answer': '(6,3),(9,3),(9,5),(54,5)', 'model answer': '(9,3),(6,3),(54,5),(9,5)'
* **6.2 Set/List -> Structural Formatting Variations**
    * *Description:* Variations in tuple, set, or list formatting, including use of braces.
    * *Example:* 'ground truth answer': '(1,2), (3,4)', 'model answer': '\{(1,2), (3,4)\}'
* **6.3 Set/List -> Element Delimiter Variations**
    * *Description:* Differences in delimiters used to separate elements (e.g., commas vs. semicolons).
    * *Example:* 'ground truth answer': '(1,2,3)', 'model answer': '(1;2;3)'
* **6.4 Set/List -> Other Set and List Variations**
    * *Description:* Other differences in set or list presentation, such as redundant parentheses.
    * *Example:* 'ground truth answer': '(1,2)', 'model answer': '((1,2))'

---

### 7. Symbolic Representation Variations

Differences in variable or constant symbols.

* **7.1 Symbolic -> Variable and Constant Choice Variations**
    * *Description:* Different letters or cases for arbitrary constants or parameters.
    * *Example:* 'ground truth answer': '...+\pi k, ...', 'model answer': '...+n \pi, ...'
* **7.2 Symbolic -> Subscript or Superscript Variations**
    * *Description:* Differences in subscript or superscript notation for variables or constants.
    * *Example:* 'ground truth answer': 'x_1, x_2', 'model answer': 'x^1, x^2'
* **7.3 Symbolic -> Custom Symbol Variations**
    * *Description:* Use of unconventional or user-defined symbols for variables or constants.
    * *Example:* 'ground truth answer': '\alpha, \beta', 'model answer': 'a, b'
* **7.4 Symbolic -> Other Symbolic Variations**
    * *Description:* Other differences in symbolic representation, such as case sensitivity.
    * *Example:* 'ground truth answer': 'P(x)', 'model answer': 'p(x)'

---

## Input

<ground_truth_answer>
{{GROUND_TRUTH_ANSWER}}
</ground_truth_answer>

<model_answer>
{{MODEL_ANSWER}}
</model_answer>

## Output

Identify the most precise equivalence or mismatch category from the taxonomy above that best characterizes the
    relationship between the ground truth answer and the model answer. Specify the primary category (required),
    and, if relevant, a secondary category (optional). Avoid selecting "Others" categories when possible.

Respond in this format, providing only the category ID and name:

<primary_category>
[ID] [Category Name] (e.g., 1.1 Formatting -> Whitespace and Spacing Issues)
</primary_category>

<second_category>
[ID] [Category Name], if applicable (e.g., 6.1 Set/List -> Order of Element Variations)
</second_category>
```

Figure 17: Prompt Template for Labeling FN Categories (Part 3)

---

**Prompt Template for Generating Synthetic FN Examples**

```
## Task Description

You are an AI assistant tasked with generating a set of mathematically equivalent answers to a given ground truth
    answer. These equivalent answers should maintain the same mathematical meaning while potentially varying in
    format, notation, or phrasing.

## Examples

Below are examples of questions with their ground truth answers, followed by equivalent answers that preserve the
    mathematical meaning.

**Example 1 (Order-Insensitive):**
<question>Determine all real values of $x$ for which $(x+8)^{4}=(2 x+16)^{2}$.</question>
<ground_truth_answer>-6,-8,-10</ground_truth_answer>

<equivalent_answer_1>-8, -10, -6</equivalent_answer_1>

**Example 2 (Latex Expression):**
<question>A bag contains 3 green balls, 4 red balls, and no other balls. Victor removes balls randomly from the bag
    , one at a time, and places them on a table. Each ball in the bag is equally likely to be chosen each time
    that he removes a ball. He stops removing balls when there are two balls of the same colour on the table. What
     is the probability that, when he stops, there is at least 1 red ball and at least 1 green ball on the table
    ?</question>
<ground_truth_answer>$\\frac{4}{7}$</ground_truth_answer>

<equivalent_answer_1>4/7</equivalent_answer_1>

**Example 3 (Variable):**
<question>If $T=x^{2}+\\frac{1}{x^{2}}$, determine the values of $b$ and $c$ so that $x^{6}+\\frac{1}{x^{6}}=T^{3}+
    b T+c$ for all non-zero real numbers $x$.</question>
<ground_truth_answer>-3,0</ground_truth_answer>
<model_answer>b=-3, c=0</model_answer>

<equivalent_answer_1>b=-3, c=0</equivalent_answer_1>
<equivalent_answer_2>b = -3, c = 0\</equivalent_answer_2>

**Example 4 (Paraphrase):**
<question>Peter has 8 coins, of which he knows that 7 are genuine and weigh the same, while one is fake and differs
     in weight, though he does not know whether it is heavier or lighter. Peter has access to a balance scale,
    which shows which side is heavier but not by how much. For each weighing, Peter must pay Vasya one of his
    coins before the weighing. If Peter pays with a genuine coin, Vasya will provide an accurate result; if a fake
     coin is used, Vasya will provide a random result. Peter wants to determine 5 genuine coins and ensure that
    none of these genuine coins are given to Vasya. Can Peter guaranteedly achieve this?</question>
<ground_truth_answer>Petya can guarantee finding 5 genuine coins.</ground_truth_answer>

<equivalent_answer_1>Yes, Peter can guarantee finding 5 genuine coins while ensuring that none of these genuine
    coins are paid to Vasya.</equivalent_answer_1>

## Input

<question>
{{QUESTION}}
</question>

<ground_truth_answer>
{{GROUND_TRUTH_ANSWER}}
</ground_truth_answer>

## Output

Please generate at least 5 mathematically equivalent answers to the ground truth answer. Each answer should be
    placed inside tags like <equivalent_answer_1>...</equivalent_answer_1>, <equivalent_answer_2>...</
    equivalent_answer_2>, etc.
```

Figure 18: Prompt Template for Generating Synthetic FN Examples

