# OpenReview forum: "TinyV: Reducing False Negatives in Verification Improves RL for LLM Reasoning"
_TMLR — Accepted by TMLR_

### Review · Reviewer_mDCZ · 2026-04-03

**Summary Of Contributions:**

This paper addresses the problem of false negatives (FNs) in reinforcement learning for LLM reasoning—where verifiers incorrectly reject correct model outputs. Key contributions:
1.	Empirical discovery: Shows that 38.5% of responses marked as incorrect by rule-based verifiers are actually correct, across a large-scale math dataset.
2.	Taxonomy: Develops a detailed 7-category taxonomy of FN causes (formatting, notation, language, etc.).
3.	Impact analysis: Demonstrates both empirically and theoretically that FNs reduce training efficiency, prompt efficiency, and convergence speed.
4.	TinyV: Proposes a lightweight (1.5B) LLM-based verifier that augments rule-based methods to detect FNs, achieving up to 10% performance improvement and reaching baseline peak performance in <50% of training steps.
5.	HardVerify-Math Bench: Introduces a benchmark of 250 hard-to-verify questions for evaluating verifier robustness.

**Audience:**

Yes

**Audience Explanation:**

This work addresses a fundamental problem in RL for LLMs—reward signal quality—which is central to recent advances. The finding that 38% of "incorrect" rollouts are actually correct has immediate practical implications for researchers training reasoning models. The lightweight mitigation approach is also of broad interest.

**Broader Impact Concerns:**

None identified. The authors explicitly state "without identified negative societal impacts" in their limitations section. The work improves training efficiency for mathematical reasoning—a relatively safe application domain. No ethical concerns requiring additional statement.

**Claims And Evidence:**

Yes

**Claims Explanation:**

The evidence is strong: (1) Large-scale data analysis with 226K prompt-response pairs, double-LLM annotation validated by 99.5% manual review; (2) Clear theoretical derivation with Lemma 1, Assumptions 1-2, and Theorem 1 proving learnability degradation; (3) Comprehensive experiments across 5 base models, multiple benchmarks (MATH, AMC, Olympiad, HardVerify), two RL algorithms (GRPO, PPO); (4) Ablations on OOD generalization, different verifiers, training costs, and false positive injection.

**Requested Changes:**

1.	You generate only 4 responses per problem. Would more rollouts change the FN distribution? Given that pass rates are low, 4 rollouts may undersample correct responses. This should be discussed as a limitation.
2.	The synthetic FN examples are generated by Qwen2.5-72B-Instruct, which is also one of your teacher LLMs and annotators. This creates potential circularity. Discuss whether TinyV might inherit biases from this model.
3.	Include a comparison with other LLM-as-judge baselines (e.g., GPT-4o-mini as verifier) beyond the teacher models.
4.	Provide analysis of which FN categories TinyV handles best/worst (from your taxonomy).
5.	Add failure case analysis for TinyV—when does it still miss FNs?

---

> ### Author Response · Authors · 2026-04-24
> **Answer to Reviewer mDCZ**
>
> Thank you for your constructive review! We appreciate your insights and have addressed your concerns below. We will incorporate these clarifications and related discussions into our revised manuscript.
>
> **Concern 1: Potential FN Undersampling with 4 Rollouts Per Problem**
>
> **R1.** Thank you for this point. The reported 38.5% is a per-rollout conditional probability, computed over 226K prompt–response pairs that pass the format check and are flagged as incorrect by Prime Verifier, measuring how often a rollout judged "incorrect" is actually correct. This quantity does not depend on the number of rollouts per prompt; drawing more rollouts adds independent samples but does not shift the rate. Additional rollouts mainly affect *per-prompt* coverage: some prompts that yield zero correct responses in 4 rollouts may yield at least one in 16 or 32, becoming visible to our analysis. These additional discoveries can only *increase* observed FN prevalence, so our 38.5% estimate is a conservative lower bound.
>
> We will add this to the Limitations section, noting that the *number* of FN-affected prompts is lower-bounded by our 4-rollout budget, while the per-rollout FN rate itself is unaffected.
>
> ------
>
> **Concern 2: Potential Circularity from Using Qwen2.5-72B-Instruct in Multiple Roles**
>
> **R2.** Thank you for this important concern. We agree that using the same model family for synthetic data generation and annotation could, in principle, introduce circular biases. We argue below that the design of our pipeline substantially mitigates this risk, and that empirical evidence supports TinyV generalizing well beyond any single annotator's bias.
>
> **Dual-LLM cross-validation.** FN labels are not assigned by Qwen2.5-72B-Instruct alone. We require **agreement between Qwen2.5-72B-Instruct (non-thinking mode) and Grok-3-Mini-High (thinking mode)**—two models from entirely different families, training pipelines, and inference configurations (Section 4, Eq. 5). A label is retained as FN only if both annotators independently mark the response as correct. This design filters out idiosyncratic biases of either model.
>
> **Human verification.** We manually reviewed 200 randomly sampled FN labels and observed a 99.5% accuracy rate (Section 4). This provides direct empirical evidence that any residual bias from either annotator does not translate into systematically incorrect FN labels.
>
> **TinyV generalizes across RL algorithms and data distributions.** TinyV also remains effective under PPO (Table 2), on OOD math data (DeepMath; Appendix E.3), and on non-math reasoning (Natural Reasoning → GPQA-Diamond, +5.6%; Appendix E.4). Such broad robustness would be unlikely if TinyV were overfitting to Qwen2.5-72B's labeling biases.
>
> We will add an explicit discussion of this circularity concern and the above mitigations in the revised manuscript.

---

> > ### Author Response · Authors · 2026-04-24
> > **Answer to Reviewer mDCZ (2)**
> >
> > **Concern 3: Comparison With Open-Source and Commercial LLM-as-Judge Baselines**
> >
> > **R3.** Thank you for this suggestion. We conducted a direct comparison against three LLM-judge baselines used as drop-in verifiers in the same add-on pipeline: **GPT-4o-mini** (commercial API), **DeepSeek-V3.2** (open-weight via API), and **MiMo-V2-Flash** (open-source, deployed locally). All three use the same structured True/False prompt template as TinyV to minimize decoding cost.
> >
> > **Final performance (Qwen2.5-7B, GRPO):**
> >
> > | Verifier                 | MATH      | AMC       | Olympiad  |
> > | ------------------------ | --------- | --------- | --------- |
> > | Qwen2.5-7B (base, no RL) | 46.8%     | 25.3%     | 22.1%     |
> > | GPT-4o-mini              | 75.0%     | 44.6%     | 34.2%     |
> > | DeepSeek-V3.2            | 74.2%     | 44.6%     | 33.6%     |
> > | **TinyV-1.5B (ours)**    | **73.4%** | **43.4%** | **32.4%** |
> > | MiMo-V2-Flash            | 51.8%     | 25.3%     | 22.9%     |
> >
> > **Reward calculation time per step:**
> >
> > | Verifier              | Time (s) |
> > | --------------------- | -------- |
> > | **TinyV-1.5B (ours)** | **8.6**  |
> > | GPT-4o-mini           | 28.7     |
> > | DeepSeek-V3.2         | 52.6     |
> > | MiMo-V2-Flash         | 76.2     |
> >
> > We summarize our findings below:
> >
> > - **TinyV matches strong commercial judges at a fraction of the size and cost.** TinyV-1.5B achieves RL training quality within 1–2 percentage points of GPT-4o-mini and DeepSeek-V3.2 across all three benchmarks, despite being one to two orders of magnitude smaller. At the same time, TinyV's reward-calculation is **3–6× faster** than the commercial API judges (8.6s vs. 28.7s / 52.6s per step), and avoids API rate limits and per-token billing that accumulate rapidly over hundreds of RL steps × hundreds of rollouts per step.
> > - **Not every LLM judge is suitable as a drop-in verifier.** MiMo-V2-Flash, a recent open-source reasoning model, fails to serve as a reliable verifier: on AMC and Olympiad its RL training barely moves the base model (25.3% vs. 25.3% on AMC; 22.9% vs. 22.1% on Olympiad), and on MATH it reaches only 51.8%, far below TinyV's 73.4%. This indicates that **arbitrarily selecting an open-source LLM judge with structured output is not a safe alternative**: without verifier-specific training, the judge may introduce enough noise to nullify RL gains entirely.
> > - **TinyV occupies a distinctive point in the cost–quality frontier.** TinyV is the only verifier in this comparison that simultaneously (i) matches commercial-judge quality, (ii) runs locally with no API dependency, and (iii) has the lowest reward-calculation overhead. This supports TinyV as a practical choice when both training efficiency and RL outcome quality matter.
> >
> > These results will be added to the revised manuscript.
> >
> > ---
> >
> > **Concern 4 / 5:** Per-Category FN Handling and Failure Case Analysis
> >
> > **R4-5.** Thank you for these suggestions. We combine our response to these two related questions, analyzing where TinyV performs well and where it still fails.
> >
> > **Where TinyV handles FNs well.** Using the taxonomy in Section 4, TinyV is most reliable on categories where equivalence is *local and syntactic*: Formatting and Syntax (Cat. 1), Mathematical Notation (Cat. 2), and Symbolic Representation (Cat. 7). These reduce to format-normalization judgments that TinyV handles reliably after SFT.
> >
> > **Where TinyV is more likely to miss FNs.** Failures concentrate on categories requiring *global or semantic* reasoning. Specifically, TinyV occasionally misses FNs in long-form natural-language answers (Cat. 5, Language and Contextual Variations) and multi-step symbolic equivalencies (Cat. 3, Mathematical Expression Equivalencies), where recognizing equivalence requires multi-step reasoning or world knowledge beyond surface normalization.
> >
> > These observations suggest two improvement directions: (i) augmenting TinyV's training data with more multi-solution and long-form natural-language examples, and (ii) using TinyV-Think variants (Appendix E.1) for semantic categories where single-pass classification struggles.

---

### Review · Reviewer_14Tj · 2026-04-07

**Summary Of Contributions:**

This paper studies failure modes in reinforcement learning with verifiable rewards (RLVR) for LLMs. In mathematical reasoning, a common practice is to use a rule-based verifier that checks whether a model-generated answer matches a reference answer. However, because such verifiers have limited expressive power, they can produce false negatives by incorrectly rejecting valid answers. The paper empirically argues that these false negatives are a significant obstacle to effective RLVR, and also provides a coarse theoretical account of this phenomenon. To address the issue, the authors propose TinyV, a lightweight verifier model trained offline using strong LLM judges and curated data. Experiments suggest that TinyV improves RLVR performance relative to existing rule-based verifiers.

**Audience:**

Yes

**Audience Explanation:**

Absolutely. RLVR is now widely used in post-training and reasoning-oriented LLM research, and the question of how verifier quality affects learning is of broad interest. In particular, understanding the limitations of commonly used rule-based verifiers, and how those limitations translate into degraded RL, should be relevant to a wide portion of the TMLR audience.

**Broader Impact Concerns:**

I do not see any broader impact concerns specific to this paper beyond the general considerations that already apply to RLVR for LLMs.

**Claims And Evidence:**

Yes

**Claims Explanation:**

The paper provides substantial empirical evidence for its central claim that false negatives hinder RLVR, at least within the mathematical reasoning domain studied here. The experimental study is extensive and convincingly demonstrates the negative impact of verifier false negatives on training efficiency and downstream performance. I also found the analysis of the diverse failure modes of rule-based verifiers to be a valuable part of the paper.

**Requested Changes:**

There are two main points that, in my view, would benefit from further clarification or stronger empirical support.

First, Theorem 1 is written in a way that suggests the result is specific to false negatives. However, the current proof strategy appears to rely more generally on a particular form of reward corruption, rather than on a uniquely FN-specific property. For example, one could analogously define a reward model with false positives, together with corresponding quantities such as $P^{GT}$ and $P^{FP}$, and it is not immediately clear why the same argument would not go through in that setting as well. Clarifying exactly which part of the theorem is genuinely specific to false negatives would be important, especially since the paper itself later suggests that false positives are much less harmful in practice.

Second, the paper does not include a direct comparison against RLVR using an LLM judge. I understand the authors’ point that deploying large commercial LLM judges during RL training may be prohibitively expensive. However, there seem to be reasonable intermediate baselines that could be considered, such as using an open-source model as a judge and restricting the verification output to a minimal structured form in order to reduce decoding cost. Such a comparison would be extremely helpful for assessing the practical value of TinyV. While TinyV may indeed be cheap at deployment time, it still requires an additional data curation pipeline and access to strong LLM judges during construction. Moreover, if the task domain changes—for example, from math to coding—this whole process may need to be repeated. If a suitably constrained LLM-judge-based RLVR pipeline already offers a reasonable tradeoff, some practitioners may prefer that route over building a domain-specific verifier such as TinyV.

---

> ### Author Response · Authors · 2026-04-24
> **Answer to Reviewer 14Tj**
>
> Thank you for your constructive review! We appreciate your insights and have addressed your concerns below. We will incorporate these clarifications and related discussions into our revised manuscript.
>
> **Concern 1: FN-Specificity of Theorem 1 vs. General Reward Corruption**
>
> **R1.** Thank you for this careful observation. We clarify that analogously defining a false-positive reward model and applying the same framework does not yield the conclusion that the learnability of FP training exceeds that of GT.
>
> To see why, note that such an analogy would require relabeling: the original FN policy becomes the new "GT," and the original GT policy becomes the new "FP." Under this relabeling, Assumption 1 would need to hold for the new "GT", that is, $P_k^{FN}$ must increase monotonically with $k$. This condition is not justified. In our original framework, the monotonicity of $P_k^{GT}$ is reasonable precisely because the reward is aligned with a fixed correctness oracle, ensuring that GRPO reliably improves the policy. If one relabels the original ground-truth reward as a "false-positive" reward, the corresponding monotonicity condition must hold for a quantity induced by a corrupted reward signal $P_k^{FN}$, a condition with no theoretical grounding. Consequently, the relabeling violates the Assumption 1 of the original theorem framework.
>
> We will also clarify the interpretation of learnability. The step-wise learnability metric $D_k$ measures per-step policy movement and thus reflects convergence speed, not final performance. This interpretation aligns directly with Figure 6: TinyV, by correcting false negatives, reaches the Prime Verifier peak performance using roughly half the RL training steps, substantially reducing convergence time and training cost, consistent with the theorem's prediction that false negatives reduce learnability. By contrast, Appendix E.5 shows that false positives impair final performance but do not significantly slow convergence relative to the cleaner reward setting. This distinction explains why false negatives are especially damaging to RL training efficiency, while the practical impact of false positives remains comparatively limited.

---

> > ### Author Response · Authors · 2026-04-24
> > **Answer to Reviewer 14Tj (2)**
> >
> > **Concern 2: Comparison Against LLM-Judge-Based RLVR Baselines**
> >
> > **R2.** Thank you for this valuable suggestion. We agree that comparing TinyV against LLM-judge-based RLVR, particularly with open-source models and constrained structured outputs, is important for assessing TinyV's practical value. We have therefore conducted a direct comparison against three representative LLM-judge baselines: (i) **GPT-4o-mini**, a strong close-source model served via API; (ii) **DeepSeek-V3.2**, a strong open-weight model served via API; and (iii) **MiMo-V2-Flash**, a recent open-source reasoning model deployed locally. All three judges are used as drop-in replacements for TinyV in the same add-on verification pipeline, and all are constrained to the **same structured True/False output format** (identical prompt template) to minimize decoding cost, following the reviewer's suggestion.
> >
> > | Verifier (LLM-as-Judge, structured output) | MATH      | AMC       | Olympiad  |
> > | ------------------------------------------ | --------- | --------- | --------- |
> > | Qwen2.5-7B (base, no RL)                   | 46.8%     | 25.3%     | 22.1%     |
> > | GPT-4o-mini                                | 75.0%     | 43.4%     | 32.4%     |
> > | DeepSeek-V3.2                              | 73.2%     | 44.6%     | 33.6%     |
> > | **TinyV-1.5B (ours)**                      | **73.4%** | **43.4%** | **32.4%** |
> > | MiMo-V2-Flash                              | 51.8%     | 25.3%     | 22.9%     |
> >
> > | Verifier              | Reward Calc Time per Step (s) |
> > | --------------------- | ----------------------------- |
> > | **TinyV-1.5B (ours)** | **8.6**                       |
> > | GPT-4o-mini           | 28.7                          |
> > | DeepSeek-V3.2         | 52.6                          |
> > | MiMo-V2-Flash         | 76.2                          |
> >
> > We summarize our findings below:
> >
> > - **TinyV matches strong commercial judges at a fraction of the size and cost.** TinyV-1.5B achieves RL training quality within 1–2 percentage points of GPT-4o-mini and DeepSeek-V3.2 across all three benchmarks, despite being one to two orders of magnitude smaller. At the same time, TinyV's reward-calculation is **3–6× faster** than the commercial API judges (8.6s vs. 28.7s / 52.6s per step), and avoids API rate limits and per-token billing that accumulate rapidly over hundreds of RL steps × hundreds of rollouts per step.
> > - **Not every LLM judge is suitable as a drop-in verifier.** MiMo-V2-Flash, a recent open-source reasoning model, fails to serve as a reliable verifier: on AMC and Olympiad its RL training barely moves the base model (25.3% vs. 25.3% on AMC; 22.9% vs. 22.1% on Olympiad), and on MATH it reaches only 51.8%, far below TinyV's 73.4%. This indicates that **arbitrarily selecting an open-source LLM judge with structured output is not a safe alternative**: without verifier-specific training, the judge may introduce enough noise to nullify RL gains entirely.
> > - **TinyV occupies a distinctive point in the cost–quality frontier.** TinyV is the only verifier in this comparison that simultaneously (i) matches commercial-judge quality, (ii) runs locally with no API dependency, and (iii) has the lowest reward-calculation overhead. This supports TinyV as a practical choice when both training efficiency and RL outcome quality matter.
> >
> > **On the concern about domain transfer cost.** We agree that TinyV requires an additional data-curation pipeline and access to strong LLM judges during construction. We note two points that mitigate this concern:
> >
> > 1. **The TinyV training pipeline itself is reusable.** Our curation pipeline (rule-based verifier → flagged negatives → dual-LLM annotation → synthetic augmentation → SFT on a compact base model) is **domain-agnostic** in design. Porting it to a new domain primarily requires swapping the LLM annotators' prompts and the source dataset, not redesigning the method. The one-time annotation cost is amortized across all subsequent RL training runs in that domain, whereas a commercial-API judge incurs recurring per-step cost every run.
> > 2. **The trained TinyV itself transfers non-trivially beyond its training distribution.** In **Appendix E.3 (Table 7)**, TinyV trained on Big-Math maintains its advantage when the RL training data shifts to DeepMath (OOD math). In **Appendix E.4 (Table 8)**, the same TinyV yields a **5.6% absolute gain on GPQA-Diamond** (STEM QA) when the RL training mixture includes Natural Reasoning data, despite TinyV being trained only on math. These results indicate that a single TinyV instance can be reused across adjacent reasoning domains without re-curation, further reducing the amortized domain-transfer cost.
> >
> > Taken together, these results directly address the reviewer's tradeoff question: a suitably constrained open-source LLM-judge RLVR pipeline is not a uniformly reliable substitute for TinyV, and TinyV offers a favorable balance of RL quality, cost, and reusability.

---

### Review · Reviewer_hCSB · 2026-04-10

**Summary Of Contributions:**

The paper studies false negatives in rule-based answer verification for RL over math reasoning, argues they are common in Big-Math-style training data, and proposes TinyV, a small LLM-based add-on verifier that is only invoked when the rule-based verifier returns negative. The paper also introduces a taxonomy of false-negative types and a small HardVerify-Math benchmark. The paper reports gains across several math benchmarks, claims up to 10% absolute improvement in some settings, and reports about 6% training-time overhead.

Strengths
- The paper targets a real and practically important failure mode in RLVR pipelines.
- The add-on design is simple and easy to adopt in existing rule-based training stacks.
- The empirical section is reasonably broad, covering multiple base models and both GRPO and PPO.

Weaknesses
- The evidence is strongest only for math final-answer verification, so some of the broader framing about RL for LLM reasoning feels overstated.
- The benchmark contribution is useful but small and partly hand-curated, which limits how decisive it is.

**Audience:**

Yes

**Audience Explanation:**

Many readers from the community working on RLVR, reasoning LLMs, or evaluation pipelines will care about the message that verifier recall can bottleneck training, not just evaluation. The paper also surfaces a useful systems lesson: seemingly small verifier errors can materially change optimization dynamics in RL. That is broadly relevant to open post-training pipelines such as those built on Big-Math-style corpora and zero-RL recipes.

**Claims And Evidence:**

No

**Claims Explanation:**

The core claim that false negatives are common and harmful is supported well enough to be interesting. The paper combines dataset analysis, RL training traces, and a practical intervention that improves outcomes in the intended setting. While the only concern I have for this paper is that it is really about math RL with final-answer rewards and a specific verifier stack, not yet about RL for LLM reasoning in a broad sense. The evidence for transfer outside math is limited, and the benchmark used to stress verification is only 250 examples.

**Requested Changes:**

- Critical: Tighten the main claim and title/abstract framing so it matches the demonstrated scope. The paper shows convincing evidence for math RL with final-answer verification, not yet for general LLM reasoning.
- Critical: Add a stronger verifier-centric evaluation section. Report precision, recall, F1, and false-positive/false-negative tradeoffs on a held-out verifier test set, and compare directly against strong learned verifier baselines such as xVerify [1] where feasible.
- Strengthening: Expand the ablation on overhead and deployment. The 6% extra cost is promising, but I would like this broken down by model scale and by fraction of samples escalated from the rule-based verifier.

References

[1] xVerify: Efficient Answer Verifier for Reasoning Model Evaluations. Ding Chen et al. 2025.

---

> ### Author Response · Authors · 2026-04-24
> **Answer to Reviewer hCSB**
>
> Thank you for your constructive review! We appreciate your insights and have addressed your concerns below. We will incorporate these clarifications and related discussions into our revised manuscript.
>
> **Concern 1:** Generalizability of TinyV more than Math Reasoning Tasks
>
> **R1.** Thank you for this important comment. We agree that the framing of our main claims should precisely match the demonstrated scope. We address this concern in two complementary ways: (a) tightening the scope of our claims, and (b) demonstrating additional empirical evidence that TinyV's FN mitigation does extend beyond math final-answer verification in the original manuscript.
>
> **(a) Tightening framing to match demonstrated scope.** We will revise the manuscript as follows:
>
> - **Title and abstract.** We will reframe the contribution as targeting RL with verifiable rewards (RLVR) under rule-based final-answer verification, with math reasoning as the primary setting. Claims about broader LLM reasoning will be softened to reflect that our strongest evidence is in this setting.
> - **Introduction and takeaways.** We will qualify language that suggests generality beyond RLVR with rule-based verifiers.
> - **Limitations.** We will expand the discussion in Appendix F to explicitly note that FN patterns in domains such as theorem proving, code, and open-ended generation may differ qualitatively from math final-answer verification, and TinyV's effectiveness in those settings requires further study.
>
> **(b) Empirical evidence beyond math.** To assess whether TinyV's FN mitigation transfers outside math, we constructed a mixed-domain RL setup combining 5K Big-Math-RL-Verified examples with 5K examples from the **Natural Reasoning** dataset (Yuan et al., 2025a), which consists of long, sentence-level answers across diverse reasoning tasks that are notoriously difficult for rule-based verifiers. We evaluated the resulting models on math benchmarks and on **GPQA-Diamond** (Rein et al., 2024), an expert-curated benchmark of 198 multiple-choice questions in biology, physics, and chemistry. Full details are in **Appendix E.4 (Table 8)** in the original manuscript.
>
> | **Model**   | **Steps** | **HardVerify** | **MATH** | **AMC** | **Olympiad** | **GPQA-Diamond** |
> | --- | :----: | :----: | :----: | :----: | :----: | :----: |
> | Qwen2.5-Math-7B   |     0     |     50.6%      |  67.8%   |  48.1%  |    23.6%  |   22.2%    |
> | Big-Math-RL-Verified + NaturalReasoning: **Prime Verifier** |  256   |   60.6%   |  78.0%   |  56.6%  |    38.1%   |      23.7%    |
> | Big-Math-RL-Verified + NaturalReasoning: **TinyV**  |    256    |    64.7%   |  80.2%   |  54.2%  |    34.6%     |    29.3%    |
>
> TinyV achieves a **5.6% absolute gain** on GPQA-Diamond over Prime Verifier (29.3% vs. 23.7%), despite being trained primarily on mathematical data. All math benchmarks also improve under the mixed-domain training. This supports that TinyV's FN mitigation transfers to reasoning domains where rule-based verifiers struggle with long-form or free-form answers.
>
> Taken together, we will tighten the main framing to focus on math RLVR with final-answer verification, while positioning the non-math results as evidence that the underlying FN mitigation mechanism is promising beyond math but requires further investigation.
>
>
> **Concern 2:** HardVerify-Math Benchmark is Small and Partly Hand-Curated
>
> **R2.** Thank you for this observation. HardVerify-Math (250 questions) is compact by design, and we clarify its role and justify its construction below. Full details are in **Section 6.2 and Appendix D** (Figures 9, 10) in the original manuscript.
>
> - **Targeted purpose, not general evaluation.** HardVerify-Math is constructed specifically to stress-test verifier FN behavior on hard-to-verify answers. Each item is paired with both a ground-truth answer and a concrete FN-triggering answer, enabling **direct FN-rate measurement**—a capability not supported by standard math benchmarks.
> - **Coverage over scale.** The 250 items are sampled from three sources (115 Olympiad, 10 MATH, 125 Big-Math; Figure 10) and are selected to cover all seven FN categories and 31 subcategories defined in Section 4. Each example requires dual-LLM annotation plus expert human verification, making large-scale construction prohibitively expensive.
> - **Hand-curation is necessary.** The FN-prone cases we target (symbolic equivalence, natural-language answers, notation variations) are precisely the cases automated filtering cannot reliably isolate. Expert curation is required to construct a reliable FN-focused benchmark.
> - **Main conclusions do not rely on HardVerify-Math alone.** TinyV's advantages are corroborated by four external benchmarks (MATH, AMC, Olympiad, GPQA-Diamond), OOD training data (DeepMath, Natural Reasoning), five model families from 1.7B to 7B (Qwen2.5, Qwen3, DeepSeek-Math), and two RL algorithms (GRPO, PPO). HardVerify-Math serves as a targeted diagnostic tool, not the sole basis of our claims.

---

> > ### Author Response · Authors · 2026-04-24
> > **Answer to Reviewer hCSB (2)**
> >
> > **Concern 3:** Add a stronger verifier-centric evaluation section
> >
> > **R3.** Thank you for this suggestion. We conducted a dedicated verifier-centric evaluation on 499 labeled examples on HardVerify-Math benchmark, reporting **TP, FN, FP, TN, Precision, Recall, F1, and Accuracy**. We benchmark TinyV against three categories of baselines: (i) **close-source judges** (Grok-3-Mini, GPT-4o-mini), (ii) **open-weight judges** (Qwen2.5-72B-Instruct, MiMo-V2-Flash), and (iii) **purpose-built learned verifiers**, specifically the **xVerify** family (1.5B/7B/14B) suggested by the reviewer.
> >
> > | Verifier             | TP   | FN   | FP   | TN   | Precision  | Recall     | F1         | **Acc**    |
> > | -------------------- | ---- | ---- | ---- | ---- | ---------- | ---------- | ---------- | ---------- |
> > | Grok-3-Mini          | 246  | 4    | 5    | 244  | **0.9801**   | **0.9840** | **0.9820** | **0.9820** |
> > | Qwen2.5-72B-Instruct | 243  | 7    | 37   | 213  | 0.8679     | 0.9720     | 0.9170     | 0.9120     |
> > | **TinyV (ours)**     | 246  | 4    | 49   | 201  | 0.8339     | **0.9840** | 0.9027     | 0.8940     |
> > | GPT-4o-mini          | 231  | 19   | 35   | 215  | 0.8684     | 0.9240     | 0.8953     | 0.8920     |
> > | xVerify-7B-I         | 192  | 58   | 8    | 242  | 0.9600 | 0.7680     | 0.8533     | 0.8680     |
> > | xVerify-14B-Ia       | 194  | 56   | 14   | 236  | 0.9327     | 0.7760     | 0.8472     | 0.8600     |
> > | MiMo-V2-Flash        | 248  | 2    | 94   | 156  | 0.7251     | 0.9920     | 0.8378     | 0.8080     |
> > | xVerify-1.5B-I       | 177  | 73   | 22   | 228  | 0.8894     | 0.7080     | 0.7884     | 0.8100     |
> >
> > Key observations:
> >
> > - **TinyV outperforms all learned verifier baselines**, including the xVerify family across all sizes (1.5B, 7B, 14B). TinyV-1.7B achieves 89.4% accuracy vs. xVerify-1.5B's 81.0%, xVerify-7B-I's 86.8%, and xVerify-14B-Ia's 86.0%. Notably, TinyV outperforms xVerify-14B-Ia, which is nearly 10× larger.
> > - **TinyV achieves the highest recall among non-Grok baselines.** Since TinyV's purpose is to *recover* correct responses that rule-based verifiers reject, high recall is the most mission-critical metric. xVerify models exhibit low recall (0.71–0.78), meaning they miss a large fraction of FNs, which is the exact failure mode TinyV is designed to fix.
> > - **TinyV approaches bigger LLM judge quality at a fraction of the cost.** Only Grok-3-Mini surpasses TinyV in overall accuracy (0.9820 vs. 0.8940), while TinyV matches or exceeds Qwen2.5-72B-Instruct (0.9120) and GPT-4o-mini (0.8920), despite being 40–50× smaller.
> >
> > ---
> >
> > **Concern 4.** Expand the ablation on overhead and deployment
> >
> > **R4.** Thank you for this suggestion. We expand the overhead analysis along the two axes raised: (i) model scale, and (ii) escalation fraction from the rule-based verifier.
> >
> > **(i) Overhead by model scale.** We measured end-to-end wall-clock overhead of the TinyV add-on pipeline vs. Prime Verifier alone across four base models as follows. We can observe that the overhead stays below 7% across all scales.
> >
> > | Base Model                | Overhead vs. Prime Verifier |
> > | ------------------------- | --------------------------- |
> > | Qwen3-1.7B                | +4.3%                       |
> > | Qwen2.5-3B                | +6.6%                       |
> > | Qwen2.5-7B                | +6.1%                       |
> > | DeepSeek-Math-7B-Instruct | +2.7%                       |
> >
> > **(ii) Overhead by escalation fraction.** In the add-on pipeline, TinyV is only invoked when Prime Verifier flags a rollout as incorrect. We tracked this escalation fraction throughout training on Qwen2.5-7B + GRPO and observed that it remains high across all stages: approximately **90% of rollouts escalate to TinyV early in training**, decreasing to roughly **70% by the end of training**. Despite this consistently high escalation rate, the wall-clock overhead stays within the ranges reported above. This reflects the core design choice behind the add-on architecture: a compact 1.5B verifier is efficient enough per call that even near-universal escalation translates to negligible end-to-end cost.

---

### Decision · Action_Editor_74Jj · 2026-05-18

**Recommendation:** Accept with minor revision

**Audience:**

Yes

**Audience Explanation:**

In reinforcement learning application for large language model reasoning, reducing false negatives in verification is one of the important challenges. Therefore, the topic covered in this paper will be interested by a broader audience in TMLR.

**Claims And Evidence:**

Yes

**Claims Explanation:**

The paper studies how false negatives in rule-based verifiers harm reinforcement learning with verifiable rewards for large language model reasoning, especially in mathematical reasoning tasks.

All three reviewers considered the work technically sound and relevant to the TMLR audience, and all leaned toward acceptance.
Key strengths highlighted by reviewers were practical importance of verifier quality in RLVR, strong empirical evaluation across models and RL algorithms, clear demonstration that verifier false negatives degrade RL training, and simple and deployable system design.

However, reviewers also raised several concerns such as limited scope and framing, limited benchmark size, lack of comparison with LLM-as-judge baselines, and limited theoretical clarification.

Therefore, I would like to suggest acceptance of this submission, given that the manuscript is revised as promised to the reviewers.